

# Microscopic derivation of Ginzburg-Landau theories for hierarchical quantum Hall states

Yoran Tournois[1*], Maria Hermanns[1,2] and Thors Hans Hansson[1]

**1** Department of Physics, Stockholm University, Stockholm, Sweden
**2** Nordita, KTH Royal Institute of Technology and Stockholm University, Stockholm, Sweden

* yoran.tournois@fysik.su.se

## Abstract

We propose a Ginzburg-Landau theory for a large and important part of the abelian quantum Hall hierarchy, including the prominently observed Jain sequences. By a generalized "flux attachment" construction we extend the Ginzburg-Landau-Chern-Simons composite boson theory to states obtained by both quasielectron and quasihole condensation, and express the corresponding wave functions as correlators in conformal field theories. This yields a precise identification of the relativistic scalar fields entering these correlators in terms of the original electron field.

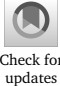
## Contents



# 1   Introduction

In spite of its more than thirty five year old history, the fractional quantum Hall effect (FQHE) [1] is still an active area of research. There are several reasons: it is the paradigmatic example of a topologically ordered state, it is a textbook example of a strongly correlated state, and it can be analyzed with a wide variety of complementary theoretical and numerical techniques. Experimentally, the FQHE is seen in a variety of systems [1–5], and there is a vast amount of data to be compared with theoretical models and numerical simulations.

The original understanding of the first observed FQH state at filling fraction $v = 1/3$ was in terms of an explicit many body wave function famously proposed by Laughlin [6]. Ever since, explicit wave functions have been very important in understanding a host of FQH liquids. There are two main strategies for constructing wave functions for more general quantum Hall liquids. The first is based on composite fermions (CF), which give a very concrete description of a large number of prominent states [7]. The second, which is more abstract, involves expressing the wave functions in terms of correlators in a two dimensional relativistic conformal field theory (CFT). The theoretical underpinning of this goes back to work by Witten [8], and it was later conjectured by Moore and Read that any QH liquid can be described in this way [9].

A radically different approach to the FQHE is the description in terms of topological quantum field theories (TQFT), which in their simplest incarnation are abelian Chern-Simons theories [10]. The TQFTs only encode the extreme low-energy and long distance properties of the QH liquids, but can be augmented by non-topological terms to describe phenomena at higher energies.

Yet another approach are the field theories of composite fermions or composite bosons (CB), which are related to the original electrons via a statistical transmutation effectuated by Chern-Simons gauge fields. Although these theories give an in principle exact microscopic description, they can in practice only be used in a mean-field framework. In this paper we concentrate on the CB theories for abelian FQH states, which are versions of the original Ginzburg-Landau-Chern-Simons theory (GLCS) [11].

In the case of the Laughlin states, the GLCS theory unifies the wave functions and the TQFT approach in that the pertinent TQFT can be derived by integrating out all dynamical degrees of freedom, while the Laughlin wave functions is retained by keeping Gaussian fluctuations around the mean-field state [12]. It is fairly straightforward to generalize this derivation to multi-component states of the Halperin type [13], where the components correspond to physically distinct particles, such as spin up/down or particles in physically separated layers.

However, there is a class of prominently observed [14] "hierarchy" states [13,15,16] in the lowest Landau level (LLL), which have so far not been possible to fit into the GLCS framework. The aim of this paper is to remedy this by constructing GLCS theories that give wave functions not only for the fully chiral part of the abelian QH hierarchy, but also for the negative Jain

series $\nu = 2/3, 3/5, \ldots$ [7]. We conjecture that our approach can be extended to the full abelian hierarchy. In the case of the Laughlin states, our derivation is essentially the same as the one by Kane *et al.* [12] (see also [17]), but it is formulated so as to give a precise relation between the non-relativistic CB quantum field and the relativistic boson fields that enter the CFT correlators giving the general hierarchy wave functions.

One might wonder if the same results can be obtained starting from a composite fermion description along the lines of Ref. [18]. However, a clear advantage of composite bosons is that it provides a natural description of states which do not correspond to integer quantum Hall states of composite fermions. A prominent example is the state at filling $\nu = 4/11$, which is discussed in section 5. It also ties to the comprehensive theoretical scheme of representative wave functions based on CFT, as we will explain below.

Technically, we will proceed by introducing generalized statistics changing phase transformations and just as in the Laughlin case, our approach relies on mean-field approximations. In addition we will, without proofs, assume that certain point-splitting regularizations of the field theory are allowed and that the related limits are well-defined. Although we believe that our derivation correctly captures important properties to the hierarchy states, the reader should be aware of these technical assumptions.

The paper is organized as follows: after giving some background material in section 2, we proceed to derive the CFT form of the Laughlin wave function from the GLCS theory in section 3 identifying the chiral boson in the CFT. We will also give an alternative derivation of the collective inter-Landau level mode. We generalize this discussion to multi-component states in section 4. The most important results are in section 5 where we introduce a generalized statistical transformation and derive the GLCS theories for both chiral and anti-chiral states in the quantum Hall hierarchy. For the latter, it is necessary to split the $K$-matrix in a difference between two positive definite parts, and to use the two terms to attach fluxes of different signs [19].

## 2 Background

The great interest in non-abelian quantum Hall states sometimes makes us forget that we are still far from a complete understanding of the experimentally much more prominent hierarchy states [14]. To underpin this contention, let us compare their status with those that we understand the best, namely the Laughlin states at filling fractions $1/m$.

Our understanding of these states is fundamentally based on the Laughlin wave function, which is the exact ground state of a special kind of short-range repulsive potentials [15, 20]. Although these potentials are singular, there is strong numerical evidence for the Coulomb ground states being in the same universality class [21]. These calculations are performed in closed geometries, so it was important that the Laughlin wave functions could also be constructed on the sphere [15] and on the torus [22].

The powerful plasma analogy introduced by Laughlin [6] can be used to establish the fractional charge and statistics of the quasiholes and quasielectrons, and the analytical results agree well with numerical calculations [23, 24]. The edges of the Laughlin liquids have been studied in detail and several different arguments lead to the conclusion that they are described by chiral Luttinger liquids [25]. The collective excitations of the Laughlin liquids were understood early on using the single mode approximation [26], and recently the quantum Hall viscosity has been calculated using several different approaches [27].

Departing from the microscopic description in terms of many-electron wave functions to various proposed effective theories, there is also a large and consistent — if not always mathematically rigorous — body of results. An effective Chern-Simons (CS) field theory can be

shown to capture all the topological properties of the Laughlin states on closed geometries and, with some very natural assumptions, also the edge states in open geometries. In its simplest version, this theory encodes both the Hall conductance and the characteristic ground state degeneracy on a torus, as well as the charge and fractional statistics of the quasiparticles. However, it was stressed by Wen that to fully characterize the topological properties of an abelian quantum Hall state, these properties are not sufficient [25].

The missing property is the density of orbital angular momentum, or "orbital spin". In a spherical geometry, this is manifested as a correction to the naive relation $N_e = \nu N_\Phi$ between the number of magnetic fluxes and the number of electrons. Instead, it becomes $N_e = \nu(N_\Phi + S)$, where the "shift" $S$ is a topological quantum number. Later, it was shown to equal the average of the orbital spin of the electrons [28], and to be proportional to the Hall viscosity, which is a non-dissipative transport coefficient.

The topological field theory and the "representative" many-body wave functions, i.e. wave functions that capture the essential low-energy properties of a phase of matter, are connected by effective field theories that were derived from the microscopic theory using various mean-field approximations. In the Ginzburg-Landau-Chern-Simons theory the electrons are turned into bosons via an Aharanov-Bohm like effect obtained by "attaching" an odd number of singular flux tubes to each particle. In a mean-field approximation these flux tubes are smeared out and their strength is chosen so as to cancel the external magnetic field, meaning that the composite bosons do not experience any net magnetic field. The resulting theory reproduces many of the properties of the Laughlin state, but it also fails in some respects [17]. It connects nicely to both of the other approaches, in that the effective CS theory can be derived from it by integrating out gapped degrees of freedom, and the Laughlin wave function can be derived using a harmonic approximation [12]. Later work showed how to properly define the GLCS theory on curved surfaces by incorporating the orbital spin of the electrons, which is related both to their cyclotron motion in the lowest Landau level and their interaction [29, 30]. To our knowledge we are the first to make a direct connection between the GLCS theory and the CFT approach to quantum Hall wave functions. Previously such a connection was only indirect in that the same wave functions were obtained by both methods. Also, previous examples of GLCS theories are few. In addition to the Laughlin states, there is work on the bosonic $\nu = 1$ nonabelian Moore-Read state, with speculations about possible extensions to bosonic Read-Rezayi states [31]. Although rather straightforward, we do not know of any work on GLCS theory for abelian multicomponent states.

A closely related approach is the Fradkin-Lopez effective fermionic field theory where the effective particles are related to the original electrons by attaching an even number of flux quanta. These composite particles experience an effective magnetic field which is nonzero, but weaker than the physical field [18]. In this description, the Laughlin states are simply a single filled Landau level of composite fermions in the effective magnetic field.

This rather complete picture of the Laughlin states should be contrasted with our much poorer understanding of the states in the abelian hierarchy. The idea of a hierarchy of quantum Hall liquids, formed by successive condensation of quasiparticles, goes back to Halperin [13] and Haldane [15]. A subset of these states — the Jain sequences at filling fractions $\nu = n/(2pn \pm 1)$ — were later very successfully described using wave functions based on the notion of composite fermions [7]. In this approach, the composite fermions are formed by attaching strength $2p$ vortices to the electrons, rather than thin flux tubes. Experiencing a reduced magnetic field, these composite fermions fill $n$ effective Landau levels, called $\Lambda$-levels. The notion of vortices was introduced by Read, who used it to derive an alternative GLCS theory for the Laughlin states [32]. This vortex-charge composite construction is also at the basis of our understanding of the (bad) metallic state observed at filling fraction $\nu = 1/2$ [7, 33].

Unlike the Laughlin states, there are no known model potentials for which the Jain states

are the exact, non-degenerate ground states [34]. There are strong heuristic reasons to believe that the quasiparticles in the Jain states have fractional charge and abelian fractional statistics [35], but lacking a simple and powerful plasma analogy the arguments are not as convincing as the case of the Laughlin quasiholes, and one has to resort to numerical simulations [36]. The CF construction has been extended to explain some hierarchy states outside the Jain series, such as the ones observed at $\nu = 4/11$ and $\nu = 5/13$, but this requires introducing interactions between the CFs and one is restricted to numerical studies on small systems [37].

Turning to the effective topological field theories, there is a general classification of abelian QH liquids based on effective CS theory [25]. In this theory, the topological data describing a quantum Hall liquid at level $n$ in the hierarchy are coded in four quantities $(K, \mathbf{t}, \mathbf{S}, \mathbf{l}^{(\alpha)})$. The $K$-matrix $K$ and the charge vector $\mathbf{t}$ together encode the filling factor and the ground state degeneracy, the (orbital) spin vector $\mathbf{S}$ determines the Hall viscosity and the response to curvature, and the $n$ vectors $\mathbf{l}^{(\alpha)}$ describe the elementary quasiparticles.

As was shown by Witten, there is a deep connection between TQFTs and conformal field theory in 1+1 dimensions [8]. More precisely, the finite-dimensional Hilbert space of a CS theory with sources can be identified with the space of conformal blocks in a CFT. It follows that the wave functions for quasiparticle excitations in QH liquids should be related to conformal blocks, but it was only through the work of Moore and Read [9] and Wen [38] that the full power of QH-CFT connection was appreciated. More precisely, Moore and Read conjectured that the holomorphic part of the actual electronic quantum Hall wave functions are conformal blocks of an Euclidean CFT and that the dynamics of the edge is described by the Minkowski version of the very same CFT. In the simplest case, both electrons and quasiparticles are described by primary fields in this CFT and their orbital spin can be identified with the conformal spin of these fields. This conjecture has been at the basis for our understanding of the non-abelian quantum Hall states, the most prominent being the Moore-Read pfaffian state which is one of the prime candidates for the observed state at $\nu = 5/2$. For a more in-depth discussion on non-abelian quantum Hall states, we refer the reader to Ref. [39].

As pointed out by Moore and Read, the CFT description is also well suited to describe the hierarchical abelian states [9]. But it was not until quite recently that this proposal was made more concrete in that explicit formulas for representative wave functions were given for the ground states and their quasiparticle excitations both on the plane [19, 40–43] and on the torus [44, 45]. In the case of the Jain series, these wave functions are identical to the ones derived using the composite fermion approach [46]. A significant technical difference between the CFT description of the Laughlin states (and also the Moore-Read pfaffian and other non-abelian states) and the hierarchy states is that for the latter, the wave functions are not single blocks of primary fields, but rather antisymmetrized sums of blocks involving descendant fields.

In conclusion, while there certainly has been much progress in understanding general hierarchical states, their more complicated structure hinders the generalization of the proofs that exist for the Laughlin states. This makes them, in a certain sense, more complicated than the non-abelian states. In particular, implementing the correct orbital spin when deriving the wave function from the effective GLCS theory has proven to be a subtle issue.

## 3 From GLCS theory to CFT correlators – the Laughlin case

We start this section by reviewing the derivation of the Laughlin wave function at $\nu = 1/k$ from the Ginzburg-Landau-Chern-Simons theory given in Refs. [12] and [17]. For details of the derivation, we refer the reader to the original references. Next, we express the composite boson wave function as a path integral and identify this expression as a correlator of vertex

operators in a bosonic two-dimensional Euclidean conformal field theory. This provides a direct connection between the composite boson field at a fixed time, and the CFT used to construct wave quantum Hall wave functions following the conjecture of Moore and Read [9].

**Notation:** In the following, we denote particle positions by $\mathbf{r} = (x, y)$ or $z = x + iy$, $\bar{z} = x - iy$, depending on which is more suitable. We will generically denote operators by hats, except when there is no risk of confusion.

## 3.1 The Ginzburg-Landau-Chern-Simons theory of the Laughlin states

We consider spin-polarized electrons in two dimensions and in the presence of a perpendicular magnetic field. The eigenvalue problem of the fermionic wave function $\Psi_F$ with the Hamiltonian

$$H_F = \frac{1}{2m} \sum_{i=1}^{N} (\mathbf{p}_i - e\mathbf{A}(\mathbf{r}_i))^2 + \sum_{i<j} V(|\mathbf{r}_i - \mathbf{r}_j|), \tag{1}$$

can be mapped onto an eigenvalue problem for bosons that have an additional statistical gauge interaction. Heuristically, the electrons are viewed as composites of bosons and flux; the dynamics is governed by a Ginzburg-Landau action for the bosons and a Chern-Simons term for the statistical gauge field. The fermionic and bosonic wave functions are related by a singular gauge transformation

$$\Psi_F(\mathbf{r}_1, \ldots, \mathbf{r}_N) = \Phi_k(\mathbf{r}_1, \ldots, \mathbf{r}_N)\Psi_B(\mathbf{r}_1, \ldots, \mathbf{r}_N)$$
$$\Phi_k(\mathbf{r}_1, \ldots, \mathbf{r}_N) = \prod_{i<j} \left( \frac{z_i - z_j}{\bar{z}_i - \bar{z}_j} \right)^{\frac{k}{2}}, \tag{2}$$

where $k$ is an odd integer, and the Hamiltonian for the bosonic eigenvalue problem reads

$$H_B = \frac{1}{2m} \sum_{i=1}^{N} (\mathbf{p}_i - e\mathbf{A}(\mathbf{r}_i) + \mathbf{a}(\mathbf{r}_i))^2 + \sum_{i<j} V\left(|\mathbf{r}_i - \mathbf{r}_j|\right). \tag{3}$$

In terms of the polar angle $2i\alpha_{ij} = \ln(z_i - z_j) - \ln(\bar{z}_i - \bar{z}_j)$ between the vectors $\mathbf{r}_i$ and $\mathbf{r}_j$, the CS gauge potential $\mathbf{a}$ is given by

$$\mathbf{a}(\mathbf{r}_i) = k\nabla \sum_{j \neq i} \alpha_{ij}, \tag{4}$$

where, in an abuse of notation, $\nabla$ denotes the gradient with respect to $\mathbf{r}_i$, whereas later on it (mostly) denotes the gradient with respect to $\mathbf{r}$.

We now turn to the second quantized Hamiltonian, introducing bosonic fields $\hat{\phi}$ and $\hat{\phi}^\dagger$. Performing a canonical transformation to the polar representation $\hat{\phi}^\dagger = \sqrt{\hat{\rho}} \, e^{-i\hat{\theta}}$ in terms of density and phase variables, it is straightforward to show that $[\hat{\rho}(\mathbf{r}), e^{-i\hat{\theta}(\mathbf{r}')}] = \delta(\mathbf{r} - \mathbf{r}')e^{-i\hat{\theta}(\mathbf{r}')}$, which yields

$$[\hat{\rho}(\mathbf{r}), \hat{\theta}(\mathbf{r}')] = i\delta(\mathbf{r} - \mathbf{r}'). \tag{5}$$

Note that, strictly speaking, the phase field $\theta$ is not Hermitian. However we are interested only in small fluctuations around a mean density, and in this case $\theta$ can be effectively treated as Hermitian, and hence $e^{i\theta}$ as being unitary. For a discussion of this point see *e.g.* [47] and references therein.

The second quantized Hamiltonian reads

$$H_B = \frac{1}{2m} \int d^2r \, |(-i\nabla - e\mathbf{A} + \mathbf{a})\hat{\phi}(\mathbf{r})|^2 + \frac{1}{2} \int d^2r \int d^2r' \delta\hat{\rho}(\mathbf{r}) V\left(|\mathbf{r} - \mathbf{r}'|\right) \delta\hat{\rho}(\mathbf{r}'), \tag{6}$$

where $\delta\hat{\rho} = \hat{\rho} - \bar{\rho}$, and $\bar{\rho}$ is the mean density. Smearing out the statistical gauge field allows for a mean-field solution $\hat{\phi} = \sqrt{\bar{\rho}}$ and $\bar{\mathbf{a}} = e\mathbf{A}$ of the GLCS equations of motion, provided $\bar{\rho}$ minimizes the potential and satisfies $2\pi k\bar{\rho} = \bar{b} = eB$. The last equality follows from the gauge constraint $2\pi k\hat{\rho} = b$ relating the density and the statistical field strength $b = \nabla \times \mathbf{a}$.

Writing $\delta\mathbf{a} = -e\mathbf{A} + \mathbf{a}$ and choosing the Coulomb gauge $\nabla \cdot \delta\mathbf{a} = 0$, we perform a derivative expansion of $H_B$, ignoring derivatives of $\hat{\rho}$ and the potential term which is proportional to $\delta\hat{\rho}^2$. At a later stage, we will reintroduce the potential term which is important for getting the correct spectrum of inter-Landau level excitations. To leading order, we find

$$
\begin{aligned}
H_B &= \frac{1}{2m}\int d^2r\,\hat{\rho}(\mathbf{r})[(\nabla\hat{\theta}(\mathbf{r})^2 + (\delta\mathbf{a})^2] \\
&\approx \frac{\bar{\rho}}{2m}\int d^2r\,\left(\hat{\theta}(\mathbf{r})(-\nabla^2)\hat{\theta}(\mathbf{r}) + \hat{\chi}(\mathbf{r})(-\nabla^2)\hat{\chi}(\mathbf{r})\right),
\end{aligned}
\tag{7}
$$

where $\delta a^i = \epsilon^{ij}\partial_j\hat{\chi}$ and $\hat{\chi}$ is related to the density fluctuation operator by

$$
2\pi k\,\delta\hat{\rho}(\mathbf{r}) = -\nabla^2\hat{\chi}(\mathbf{r}).
\tag{8}
$$

Subtracting an ultraviolet divergent constant (which amounts to normal ordering), the Hamiltonian can be rewritten as

$$
H_B = \omega_c\int d^2r\,\hat{a}^{\dagger}(\mathbf{r})\hat{a}(\mathbf{r})
\tag{9}
$$

where $\omega_c = \frac{eB}{m}$ is the cyclotron frequency. To obtain the lowering and raising operators

$$
\begin{aligned}
\hat{a}(\mathbf{r}) &= \sqrt{\frac{1}{\pi k}}\,\bar{\partial}\left(\hat{\theta}(\mathbf{r}) - i\hat{\chi}(\mathbf{r})\right) \\
\hat{a}^{\dagger}(\mathbf{r}) &= \sqrt{\frac{1}{\pi k}}\,\partial\left(\hat{\theta}(\mathbf{r}) + i\hat{\chi}(\mathbf{r})\right),
\end{aligned}
\tag{10}
$$

we expressed $\nabla^2 = 4\partial\bar{\partial}$ in terms of the holomorphic and anti-holomorphic derivatives $\partial \equiv \partial_z$ and $\bar{\partial} \equiv \partial_{\bar{z}}$. They obey the standard commutation relations

$$
[\hat{a}(\mathbf{r}), \hat{a}^{\dagger}(\mathbf{r}')] = \delta(\mathbf{r} - \mathbf{r}').
\tag{11}
$$

In the $\rho$-representation, where $\hat{\rho}(\mathbf{r})$ acts multiplicatively and $\hat{\theta}(\mathbf{r}) = -i\frac{\delta}{\delta\rho(\mathbf{r})}$, the (unnormalized) bosonic ground state wave functional reads

$$
\langle\rho(\mathbf{r})|\Psi_B\rangle = e^{\frac{k}{2}\int d^2r\int d^2r'\rho(\mathbf{r})\ln|\mathbf{r}-\mathbf{r}'|\rho(\mathbf{r}')}e^{-k\bar{\rho}\int d^2r\int d^2r'\rho(\mathbf{r}')\ln|\mathbf{r}-\mathbf{r}'|}.
\tag{12}
$$

By substituting the expression for the density $\rho(\mathbf{r}) = \sum_{i=1}^N\delta(\mathbf{r} - \mathbf{r}_i)$, we get the composite boson wave function,

$$
\Psi_B(\mathbf{r}_1, \ldots, \mathbf{r}_N) = \prod_{i<j}|\mathbf{r}_i - \mathbf{r}_j|^k e^{-\sum_i\frac{|\mathbf{r}_i|^2}{4\ell^2}},
\tag{13}
$$

which is nothing but the absolute value of the ordinary Laughlin wave function. Here, we assume that the short distance singularities in Eq. (12) are regularized. The excited states follow by repeated action with the operators $\hat{a}^{\dagger}(\mathbf{r})$ on $|\Psi_B\rangle$ in Eq. (12). Note that although the ground state is unique, the excited states are massively degenerate. We will return to this point below in section 3.3.

## 3.2 The Laughlin wave function as a CFT correlator

The basic insight that will allow us to connect the GLCS theory to the CFT expression for the wave function is to use the $\theta$-representation of the ground state wave functional. This is obtained in a similar way to (12); by writing $\hat{\chi}$ in terms of $\delta\hat{\rho} = \hat{\rho} - \bar{\rho}$ and using $\hat{\rho} = i\frac{\delta}{\delta\theta}$, we find the solution[1]

$$\langle \theta(\mathbf{r}) | \Psi_B \rangle = e^{\frac{1}{4\pi k} \int d^2 r\, \theta(\mathbf{r}) \nabla^2 \theta(\mathbf{r})} e^{-i\bar{\rho} \int d^2 r\, \theta(\mathbf{r})}. \tag{14}$$

We now rewrite the composite boson wave functional Eq. (12) as a two-dimensional path integral by inserting a resolution of unity, and use the expression $\langle \rho(\mathbf{r}) | \theta(\mathbf{r}) \rangle = = \exp\left[ i \int d^2 r\, \rho(\mathbf{r}) \theta(\mathbf{r}) \right] = e^{i\theta(\mathbf{r}_1)} \cdots e^{i\theta(\mathbf{r}_N)}$ for the overlap between density and phase eigenstates. We find[2]

$$\begin{aligned} \Psi_B(\mathbf{r}_1, \ldots, \mathbf{r}_N) &= \int \mathcal{D}[\theta(\mathbf{r})] \, \langle \rho(\mathbf{r}) | \theta(\mathbf{r}) \rangle \, \langle \theta(\mathbf{r}) | \Psi_B \rangle \\ &= \left\langle e^{i\theta(\mathbf{r}_1)} \cdots e^{i\theta(\mathbf{r}_N)} e^{-i\bar{\rho} \int d^2 r\, \theta(\mathbf{r})} \right\rangle, \end{aligned} \tag{15}$$

where $\langle \cdots \rangle$ is a correlator taken with respect to the action $S[\theta] = \frac{1}{4\pi k} \int d^2 r\, \nabla\theta(\mathbf{r}) \cdot \nabla\theta(\mathbf{r})$. This Gaussian path integral is evaluated straightforwardly, and yields precisely the expression (13) for the composite boson wave function.

Since $S$ is the action of a conformal field theory, Eq. (15) is the path integral expression for the CFT correlator

$$\left\langle : e^{i\hat{\theta}(\mathbf{r}_1)} : \cdots : e^{i\hat{\theta}(\mathbf{r}_N)} :: e^{-i\bar{\rho} \int d^2 r\, \hat{\theta}(\mathbf{r})} : \right\rangle = \prod_{i<j} |\mathbf{r}_i - \mathbf{r}_j|^k e^{-\sum_i \frac{|\mathbf{r}_i|^2}{4\ell^2}}. \tag{16}$$

Here $\langle AB \cdots \rangle$ is short for the vacuum expectation value of the radially ordered product of the (normal ordered) operators $AB \cdots$. This identity, which follows from applying Wick's theorem, expresses the absolute value of the Laughlin wave function as a CFT correlator, and is an important intermediate result of this paper.

To recover the full fermionic wave function $\Psi_F$, we must reintroduce the phase factor $\Phi_k$ in Eq. (2). Note that we do *not* get the wave function by simply extracting the holomorphic conformal block from (16). In the Laughlin case it differs just by a square root, but in the general case discussed below this is not true, and it is crucial to include the phase factor from the statistics changing transformation.

Instead, we proceed by factoring the correlator of the bosonic wave function Eq. (16) into a holomorphic and an anti-holomorphic block. Additionally, we renormalize $\hat{\theta}$ with a factor $\sqrt{k}$ such that it has the two-point function $\langle \hat{\theta}(\mathbf{r}) \hat{\theta}(\mathbf{r}') \rangle = -\ln|\mathbf{r} - \mathbf{r}'|$. We find

$$\left\langle : e^{i\sqrt{k}\hat{\theta}(\mathbf{r}_1)} : \cdots : e^{i\sqrt{k}\hat{\theta}(\mathbf{r}_N)} :: e^{-i\sqrt{k}\bar{\rho} \int d^2 r\, \hat{\theta}(\mathbf{r})} : \right\rangle = \left| \left\langle : e^{i\sqrt{k}\hat{\theta}(z_1)} : \cdots : e^{i\sqrt{k}\hat{\theta}(z_N)} : \mathcal{O}_b[\hat{\theta}(z)] \right\rangle \right|^2. \tag{17}$$

Here, the holomorphic field $\hat{\theta}(z)$ has the two point function $\langle \hat{\theta}(z) \hat{\theta}(z') \rangle = -\frac{1}{2} \ln(z - z')$, and $\mathcal{O}_b[\hat{\theta}(z)] =: e^{-i\sqrt{k}\bar{\rho} \int d^2 r\, \hat{\theta}(z)} :$ is the background charge operator. We introduce an auxiliary

---

[1]Note that it is also possible to keep $\delta\hat{\rho}$ and replace it with $i\frac{\delta}{\delta\theta}$. This gives a slightly different $\theta$-representation consisting only of the first term in (14). Because the second term is a background charge term from the CFT perspective, one might worry that the resulting correlator is not charge neutral in this modified $\theta$-representation. However, in this description the background charge results from the overlap $\langle \delta\rho | \theta \rangle$, and yields the same result as in (15).

[2] Alternatively we can write $\langle \mathbf{r}_1, \ldots, \mathbf{r}_N | = \langle 0 | \hat{\phi}(\mathbf{r}_1) \cdots \hat{\phi}(\mathbf{r}_N)$ and calculate the overlap with $|\theta\rangle$ by using the polar decomposition $e^{i\theta}\sqrt{\rho}$ of the annihilation operator. This yields the same result as in the text up to a (singular) normalization.

field $\hat{\varphi}$, also normalized as $\left\langle \hat{\varphi}(z)\hat{\varphi}(z')\right\rangle = -\frac{1}{2}\ln(z-z')$, and express the phase factor $\Phi_k$ as

$$\Phi_k(\mathbf{r}_1,\dots,\mathbf{r}_N) = \frac{\left\langle W(z_1)\cdots W(z_N)\mathcal{O}_b[\varphi(z)]\right\rangle}{\left\langle \bar{W}(\bar{z}_1)\cdots \bar{W}(\bar{z}_N)\mathcal{O}_b[\bar{\varphi}(\bar{z})]\right\rangle}. \tag{18}$$

The associated vertex operators read $W(z) =: e^{i\sqrt{k}\hat{\varphi}(z)}:$, and a similar expression for $\bar{W}(\bar{z})$ in terms of $\hat{\bar{\varphi}}$.

Multiplying the bosonic wave function (17) with the phase factor $\Phi_k$ in Eq. (18), the anti-holomorphic factors cancel. Therefore, the fermionic wave function reads

$$\begin{aligned} \Psi_F(z_1,\dots,z_N) &= \langle 0| :e^{i\sqrt{k}\hat{\phi}(z_1)}:\cdots:e^{i\sqrt{k}\hat{\phi}(z_N)}: \mathcal{O}_b[\hat{\phi}(z)] \,|0\rangle \\ &= \prod_{i<j}(z_i-z_j)^k e^{-\frac{1}{4\ell^2}\sum_i |z_i|^2}, \end{aligned} \tag{19}$$

where $\hat{\phi} = \hat{\theta} + \hat{\varphi}$ obeys $\left\langle \hat{\phi}(z)\hat{\phi}(z')\right\rangle = -\ln(z-z')$. The vacuum expectation value is taken in the product CFT of the free boson CFTs for $\theta$ and $\varphi$. Introducing the vertex operators,

$$V(z) =: e^{i\sqrt{k}\hat{\phi}(z)}:, \tag{20}$$

we write the wave function as

$$\Psi_F(z_1,\dots,z_N) = \left\langle V(z_1)\cdots V(z_N)\mathcal{O}_b[\phi(z)]\right\rangle. \tag{21}$$

The identification of the holomorphic field $\phi$ in terms of the phase field $\theta$, leading to the well-known expression (21) of the fermionic Laughlin wave function as a CFT correlator, is one of the main results of this paper.

To summarize, we employed the phase representation to express the composite boson wave function as a path integral expression and subsequently as a vacuum expectation value in the free boson CFT of the phase field $\theta$. Multiplying with the phase factor $\Phi_k$, we wrote the fermionic wave function as a holomorphic correlator in the field $\hat{\phi}(z) = \hat{\theta}(z) + \hat{\varphi}(z)$. These steps will, mutatis mutandis, be repeated in more complicated cases in the following sections.

A comment on the background charge operators in (17), (18) and (21) is in order. These operators ensure charge neutrality in the correlators, and serve to reproduce the multiplicative Gaussian factor $\exp(-\frac{1}{4\ell^2}\sum_i |z_i|^2)$ which is characteristic for the Landau level wave functions. In each holomorphic or anti-holomorphic correlator, the operator $\mathcal{O}_b$ reproduces the square root of the Gaussian factor, times a singular phase. In the bosonic wave function (17), these singular phases cancel and the final result is well-defined. However, the singular phase does appear in the fermionic wave function after multiplying with the phase factor $\Phi_k$. For details on how to regularize such phases we refer to Appendix A of Ref. [40].

### 3.3 Excited states

Two kinds of excitations are naturally described using composite bosons: the quasiholes and the inter-Landau level excitation related to the Kohn mode. Following Laughlin, the former amounts to introducing a unit strength vortex, while the latter amounts to acting on the ground state with the creation operators $\hat{a}^\dagger(\mathbf{r})$ (10). Another important collective intra-Landau level excitation is the magnetoroton, first described by Girvin, MacDonald and Platzman using the so called single mode approximation [26]. This is harder to describe in the composite boson approach, but it can be incorporated in a more elaborate framework where the composite boson theory is coupled to nematic order parameter [48]. This theory is also closely related to theories based on dynamical metrics [49,50].

### 3.3.1 Charged anyonic excitations

We consider the $\nu = 1/k$ Laughlin state with thin unit flux tubes inserted at the positions $\eta_1, \ldots, \eta_M$. The phase factor (2) is modified to

$$\Phi_k(\mathbf{r}_1, \ldots, \mathbf{r}_N) \to \Phi_k(\mathbf{r}_1, \ldots, \mathbf{r}_N) \prod_{i,a} \left( \frac{z_i - \eta_a}{\bar{z}_i - \bar{\eta}_a} \right)^{\frac{1}{2}}, \tag{22}$$

where $\eta, \bar{\eta}$ are the holomorphic and anti-holomorphic coordinates of $\eta$. The Chern-Simons gauge field $\mathbf{a}$ in Eq. (4) is then given by

$$\mathbf{a}(\mathbf{r}_i) = k\nabla \Big( \sum_{j \neq i} \alpha_{ij} + \frac{1}{k} \sum_a \alpha_{ia} \Big), \tag{23}$$

where $\alpha_{ia}$ denotes the angle between $\mathbf{r}_i$ and $\eta_a$. This amounts to a shift $(1/k)\sum_a \delta(\mathbf{r} - \eta_a)$ in the statistical gauge field strength, which modifies the Hamiltonian (7) by the shift

$$\delta\hat{\rho}(\mathbf{r}) \to \delta\hat{\rho}(\mathbf{r}) + \frac{1}{k} \sum_{a=1}^{M} \delta(\mathbf{r} - \eta_a) \tag{24}$$

in the density fluctuation (or equivalently in $\hat{\rho}$). Such a shift is implemented by the operator

$$\hat{T}_{\eta_1, \ldots, \eta_M} = e^{i \int d^2r \, \hat{\theta}(\mathbf{r}) \frac{1}{k} \sum_{a=1}^{M} \delta(\mathbf{r} - \eta_a)} = e^{\frac{i}{k} \sum_{a=1}^{M} \hat{\theta}(\eta_a)} \tag{25}$$

in the $\theta$-representation, as $\theta$ is canonically conjugate to the density operator. Note that $\theta$ denotes the original $\theta$-field prior to the rescaling. With this, the wave functional (14) becomes

$$\left\langle \theta(\mathbf{r}) | \hat{T}_{\eta_1, \ldots, \eta_M} | \Psi_B \right\rangle = e^{\frac{i}{k} \sum_{a=1}^{M} \theta(\eta_a)} e^{-i\bar{\rho} \int d^2r \, \theta(\mathbf{r})} e^{\frac{1}{4\pi k} \int d^2r \, \theta(\mathbf{r}) \nabla^2 \theta(\mathbf{r})}. \tag{26}$$

Rescaling the field $\hat{\theta}$ by $\sqrt{k}$ and performing the same steps that led from (14) to the final result (21), we get the standard CFT expression for the fermionic wave function with $M$ holes

$$\Psi_F(z_1, \ldots, z_N; \eta_1, \ldots, \eta_M) = \Big\langle \prod_{a=1}^{M} H(\eta_a) \prod_{i=1}^{N} V(z_i) \mathcal{O}_b[\phi(z)] \Big\rangle, \tag{27}$$

where the quasihole operator is given by

$$H(\eta) =: e^{\frac{i}{\sqrt{k}} \hat{\phi}(\eta)} :. \tag{28}$$

Note that also here using the $\theta$-representation for the wave functional is important in order to establish the relation to the CFT expression.

If we were instead to consider a state with "reverse" flux tubes, we would act with the operator $\hat{T}^{-1}$, which amounts to inserting the inverse quasihole operator $H^{-1}$ in the correlator. Although this does have the correct topological properties describing quasielectrons, the resulting wave function is not an acceptable LLL wave function as explained in Refs. [51, 52].

### 3.3.2 Neutral, inter-Landau level excitations

Since (7) is a collection of harmonic oscillators, neutral excitations are obtained by acting with the raising operators $a^\dagger(\mathbf{r})$ in Eq. (10). The energy of one such excitation is $\hbar\omega_c$ so this clearly describes excitations to higher Landau levels. To calculate the dispersion relation for this so-called Kohn mode, it is advantageous to go to momentum space and use

$$\hat{a}^\dagger(\mathbf{q}) = \int d^2r \, e^{-i\mathbf{q}\cdot\mathbf{r}} \hat{a}^\dagger(\mathbf{r}), \tag{29}$$

satisfying $[\hat{a}(\mathbf{q}), \hat{a}^\dagger(\mathbf{p})] = (2\pi)^2 \delta(\mathbf{p} - \mathbf{q})$. Defining the state $\left|\Psi_\mathbf{q}\right\rangle = \hat{a}^\dagger(\mathbf{q}) \left|\Psi_B\right\rangle$, we calculate the shift in energy $\Delta E_\mathbf{q} = \left\langle \Psi_\mathbf{q}|H_I|\Psi_\mathbf{q}\right\rangle / \left\langle \Psi_\mathbf{q}|\Psi_\mathbf{q}\right\rangle$. Since the density fluctuation operator is

$$\delta\hat{\rho}(\mathbf{q}) = \frac{i}{2\sqrt{\pi k}}\left(\bar{q}\,\hat{a}^\dagger(-\mathbf{q}) + q\,\hat{a}(-\mathbf{q})\right), \tag{30}$$

the (normal ordered) interaction Hamiltonian $H_I = \frac{1}{2}\int d^2r \int d^2r' \delta\hat{\rho}(\mathbf{r})V(|\mathbf{r}-\mathbf{r}'|)\delta\hat{\rho}(\mathbf{r}')$ becomes

$$H_I = \frac{1}{4\pi k}q^2 V(\mathbf{q})\hat{a}^\dagger(\mathbf{q})\hat{a}(\mathbf{q}). \tag{31}$$

To obtain (31) we ignored terms that do not contribute to the shift in energy. Assuming a normalized state $|\Psi_B\rangle$, the excited states are normalized as

$$\left\langle \Psi_\mathbf{q}|\Psi_\mathbf{q}\right\rangle = (2\pi)^2\delta(0), \tag{32}$$

where the singular factor should be interpreted as the (infinite) area $\int d^2r$. To lowest order in perturbation theory the energy shift is given by

$$\left\langle \Psi_\mathbf{q}|H_I|\Psi_\mathbf{q}\right\rangle = (2\pi)^2\delta(0)\frac{1}{4\pi k}q^2 V(q) \tag{33}$$

and, using $2\pi k\bar{\rho}\ell^2 = 1$, yields the known result [53].

$$\Delta E_\mathbf{q} = \frac{\bar{\rho}}{2eB}q^2 V(q). \tag{34}$$

## 4 The multi-component case

The GLCS theory for multi-component quantum Hall states [13, 54] is a straightforward generalization of the results in the previous section. It is of direct experimental relevance for electrons in different spin states and/or in two different layers, corresponding to a two- or four-component liquid. We consider the general case with $n$ components, characterized by an $n \times n$ $K$-matrix encoding the correlations between the various components.

Since the particles are distinguishable, we can perform a different phase transformation for different components. For a system with $N$ particles divided in $n$ subsets $M_\alpha$ with $N_\alpha$ particles in each set, we generalize the phase transformation (2) to[3]

$$\Phi_K(\mathbf{r}_1,\ldots,\mathbf{r}_N) = \prod_{\alpha \leq \beta}^{n} \prod_{\substack{i\in M_\alpha \\ j\in M_\beta}} \left(\frac{z_i - z_j}{\bar{z}_i - \bar{z}_j}\right)^{\frac{1}{2}K_{\alpha\beta}}, \tag{35}$$

giving rise to the gauge fields $a^\alpha(\mathbf{r}_i) = \sum_\beta K_{\alpha\beta} \sum_{j\neq i} \nabla\alpha_{ij}$ that couple to the composite bosons. As in the single component case one can find a mean-field solution, and the total and partial densities are given by

$$\bar{\rho} = \sum_\alpha \bar{\rho}_\alpha = \frac{eB}{2\pi}\sum_{\alpha,\beta} K_{\alpha\beta}^{-1}. \tag{36}$$

This generalizes the relation $2\pi k\bar{\rho} = eB$ in the one-component case. In the polar representation $\hat{\phi}_\alpha = e^{i\hat{\theta}_\alpha}\sqrt{\hat{\rho}_\alpha}$, the generalization of the Hamiltonian (7) becomes

$$
\begin{aligned}
H_B &= \sum_\alpha \frac{1}{2m}\int d^2r |(-i\nabla - e\mathbf{A} + \mathbf{a}^\alpha)\hat{\phi}_\alpha|^2 + \frac{1}{2}\int d^2r \int d^2r' \delta\hat{\rho}(\mathbf{r})V(|\mathbf{r}-\mathbf{r}'|)\delta\hat{\rho}(\mathbf{r}') \\
&\approx \sum_\alpha \frac{\bar{\rho}_\alpha}{2m}\int d^2r \left[\hat{\theta}_\alpha(\mathbf{r})(-\nabla^2)\hat{\theta}_\alpha(\mathbf{r}) + \hat{\chi}_\alpha(\mathbf{r})(-\nabla^2)\hat{\chi}_\alpha(\mathbf{r})\right],
\end{aligned}
\tag{37}
$$

---

[3]Here we also assume that for $\alpha = \beta$, the product is over $i < j$

where $\hat{\chi}_\alpha$ is related to the partial densities in direct analogy with (8)

$$2\pi K_{\alpha\beta}\delta\hat{\rho}_\beta(\mathbf{r}) = -\nabla^2\hat{\chi}_\alpha(\mathbf{r}). \tag{38}$$

Following the same steps as before, it is straightforward to verify that the ground state wave functional in the $\theta$-representation reads

$$\langle\theta(\mathbf{r})|\Psi_B\rangle = e^{\frac{1}{4\pi}\int d^2r\,\theta_\alpha(\mathbf{r})K_{\alpha\beta}^{-1}\nabla^2\theta_\beta(\mathbf{r})}e^{-i\bar{\rho}_\alpha\int d^2r\,\theta_\alpha(\mathbf{r})}, \tag{39}$$

where $\theta$ denotes the collection of $n$ phase fields with two-point function $\langle\theta_\alpha(\mathbf{r})\theta_\beta(\mathbf{r}')\rangle = -K_{\alpha\beta}\ln|\mathbf{r}-\mathbf{r}'|$. In the case that the $K$-matrix has rank $n$, we can write it as

$$K = QQ^T, \tag{40}$$

in terms of an $n\times n$ matrix $Q$ (in general, $Q$ is $n\times k$, $k\geq n$, see the comment at the end of section 5.2.2.). We then perform the change of variables

$$\theta_\alpha(\mathbf{r}) \to Q_{\alpha\beta}\theta_\beta(\mathbf{r}), \tag{41}$$

which generalizes the rescaling in the previous section. The rescaled fields are independent, and the two-dimensional path integral expression for the composite boson wave function becomes (up to a normalization)

$$\Psi_B(\mathbf{r}_1,\ldots,\mathbf{r}_N) = \int \mathcal{D}[\theta]\prod_{\alpha=1}^{n}\prod_{i\in M_\alpha}[e^{iQ_{\alpha\beta}\theta_\beta(\mathbf{r}_i)}]e^{-i\bar{\rho}_\alpha Q_{\alpha\beta}\int d^2r\,\theta_\beta(\mathbf{r})}e^{\frac{1}{4\pi}\int d^2r\,\theta_\alpha(\mathbf{r})\nabla^2\theta_\alpha(\mathbf{r})}. \tag{42}$$

We view this as the path integral version of the correlator

$$\Psi_B(\mathbf{r}_1,\ldots,\mathbf{r}_N) = \Big\langle\prod_{\alpha}\prod_{i\in M_\alpha}e^{iQ_{\alpha\beta}\hat{\theta}_\beta(\mathbf{r})}e^{-i\bar{\rho}_\alpha Q_{\alpha\beta}\int d^2r\,\hat{\theta}_\beta(\mathbf{r})}\Big\rangle. \tag{43}$$

Since the new fields are independent, we can factor the full correlator into correlators for each individual field $\theta_\alpha$, which subsequently factors into holomorphic and anti-holomorphic parts as in the previous section. To obtain the fermionic wave function we multiply with the phase factor $\Phi_K$, written as

$$\Phi_K(\mathbf{r}_1,\ldots,\mathbf{r}_N) = \frac{\big\langle\prod_{\alpha,i}e^{iQ_{\alpha\beta}\hat{\varphi}_\beta(z_i)}\mathcal{O}_b[\hat{\varphi}]\big\rangle}{\big\langle\prod_{\alpha,i}e^{iQ_{\alpha\beta}\hat{\bar{\varphi}}_\beta(\bar{z}_i)}\mathcal{O}_b[\hat{\bar{\varphi}}]\big\rangle}. \tag{44}$$

The resulting expression is again a correlator

$$\Psi_F(z_1,\ldots,z_N) = \Big\langle\prod_{\alpha=1}^{n}\prod_{i\in M_\alpha}V_\alpha(z_i)\mathcal{O}_b[\hat{\phi}]\Big\rangle, \tag{45}$$

where the vertex operators are given by

$$V_\alpha(z) = e^{iQ_{\alpha\beta}\hat{\phi}_\beta(z)}. \tag{46}$$

The fields $\hat{\phi}_\alpha = \hat{\theta}_\alpha + \hat{\varphi}_\alpha$ obey $\langle\hat{\phi}_a(z)\hat{\phi}_b(z')\rangle = -\delta_{ab}\ln(z-z')$, and the vacuum expectation value is taken with respect to the product of the free boson CFTs for each of the $\hat{\phi}_\alpha$.

# 5 Orbital spin and the hierarchy

## 5.1 General discussion

A mean-field approximation often proceeds by first introducing redundant auxiliary variables and subsequently eliminating all or many of the original degrees of freedom. In doing so, one obtains an effective low-energy theory in terms of the new variables. This is most easily done using path integrals, where the auxiliary variables typically are introduced via a Hubbard-Stratonovich transformation, and the elimination proceeds through integration over the microscopic degrees of freedom. There is a large freedom in choosing the auxiliary fields, and different choices will give different effective field theories. The goal is to find a formulation where the effective theory captures the most important infrared properties already at the classical, or mean-field, level. In conventional systems, this amounts to finding the correct pattern of spontaneous symmetry breaking, while in topologically ordered systems it means finding the correct values for topologically protected quantities such as fractional charges. In but the simplest cases, the actual choice of mean-field is guided by a mixture of theoretical intuition and phenomenological input. In favorable cases one can use numerical calculations to determine which of several competing mean-field states is energetically favored, and also get an improved description by a perturbative expansion around the mean-field.

The procedure used in Section 3 to derive the GLCS theory is subtle, in that the reformulation of the theory amounts to performing a "singular gauge transformation" rather than introducing an auxiliary dynamical variable. This notion, although commonly used, is easily misunderstood. A regular gauge transformation has no physical meaning, while a singular transformation typically does. This is most easily seen by considering the introduction of an infinitesimally thin, but fractional, flux tube at some fixed position. If we consider a particle with unit charge, and require the wave function to be single valued, the spectrum will change. If one allows for a simultaneous change of the Hamiltonian and the periodicity condition on the wave function, the spectrum can be made invariant with exception of the s-wave, since the wave function is forced to vanish at the position of the flux tube. A similar logic applies to the statistics changing singular gauge transformations in Eq. (35) considered in the last section. Again, these are well-defined only if the wave function vanish at the point of coincidence, i.e. for $\mathbf{r}_i = \mathbf{r}_j$. For even $k$, this is always satisfied since the transformed wave function is fermionic, but for odd $k$, as in the derivation of GLCS theory, this amounts to the extra constraint that the bosons of the transformed theory must have "hard cores". The wave functions derived in the previous sections do describe hard core bosons, so in this sense the calculation is self-consistent.

In this section we present a formalism, based on generalized flux attachment procedures, that allows us to endow the electrons with arbitrary orbital spin. Using this we derive a GLCS theory for hierarchy states, which will reproduce the CFT expressions for the wave functions.

The simplest and most prominent hierarchy states are those in the leading positive Jain series, at $\nu = \frac{n}{2n+1}$, corresponding to taking $p = 1$ in the general formula $\nu = \frac{n}{2pn+1}$. In terms of composite fermions, a description of the orbital spin is automatically included, as composite fermions in different $\Lambda$-levels have different orbital spins. In our proposed GLCS theory, orbital spin is included by generalizing the flux attachment to allow for a separation of flux and charge. We show that, similar to the Laughlin and multi-component wave functions, the wave functions in the positive Jain series may be obtained as a CFT correlator from the GLCS theory. We proceed by generalizing this GLCS theory to the full chiral hierarchy, which differs from the positive Jain series in that the different $\Lambda$-levels need not have identical correlations. We then proceed to the leading negative Jain sequence $\nu = \frac{n}{2n-1}$, where the flux attachment procedure is further generalized to allow for reverse flux attachment. Finally we offer a conjecture about

the full hierarchy.

Before proceeding with the technical details, we want to note an important point. Our aim is to derive representative wave functions starting from the GLCS theory and a given set of topological data. Our construction gives us a wave function with the wanted topological properties, but that does not imply that it will be the ground state of a realistic Hamiltonian nor that the overlap with a numerically obtained ground state using Coulomb interaction is particularly high. These latter questions need to be addressed separately using numerical methods. Note also that there are various ways to modify a wave function without changing its topological properties, thus making these wave functions variational. A particular method is briefly discussed at the end of Sect. 5.3. There is also an early discussion in Ref. [9], and an approach based on excitons is proposed in chapter 5 of Ref. [55].

## 5.2 From multi-component states to the chiral hierarchy

The chiral hierarchy forms a subset of hierarchy states resulting from the successive condensation of quasielectrons only. As mentioned above, to derive a GLCS theory for these states requires the introduction of orbital spin. We first illustrate the procedure by working out the simplest example in detail, which is the first member of the positive Jain sequence at $\nu = 2/5$.

### 5.2.1 The $\nu = 2/5$ state

The $\nu = 2/5$ state is the most prominent state in the chiral hierarchy, and it can be viewed as the first daughter state of the Laughlin $\nu = 1/3$ resulting from the condensation of quasielectrons, or in parlance of composite fermions as the state where two CF Landau levels are filled, attaching two quantized vortices to each electron. We begin by comparing this state to the two-component Halperin $(3, 3, 2)$ state. Both are described by the $K$-matrix

$$K = \begin{pmatrix} 3 & 2 \\ 2 & 3 \end{pmatrix}, \tag{47}$$

but the $\nu = 2/5$ state cannot be obtained from the $(3, 3, 2)$ state by antisymmetrizing, since the latter state treats the two components symmetrically. What is missing is the orbital spin [10], which distinguishes the two groups for the $\nu = 2/5$ state.

Thus, in order to derive the proper GLCS theory, we incorporate orbital spin in the bosonic formulation by generalizing the flux attachment mechanism so as to allow for a spatial separation between the charge and the flux. To this end, we note that in the polar representation of the bosonic field the operator $e^{i\hat{\theta}(\mathbf{r})}$ creates a point charge when acting on a density eigenstate $|\rho\rangle$, while the operator $\hat{\rho}$ creates a point current when acting on a current eigenstate $|\mathbf{J}\rangle$. That is,

$$\hat{\rho}(\mathbf{r})e^{i\hat{\theta}(\mathbf{r}')}|\rho\rangle = (\rho(\mathbf{r}) + \delta(\mathbf{r} - \mathbf{r}'))e^{i\hat{\theta}(\mathbf{r}')}|\rho\rangle$$

$$\hat{J}_i(\mathbf{r})\hat{\rho}(\mathbf{r}')|\mathbf{J}\rangle = (J_i(\mathbf{r}) - \frac{i}{m}\partial_i\delta(\mathbf{r} - \mathbf{r}'))\hat{\rho}(\mathbf{r}')|\mathbf{J}\rangle,$$

with $\hat{J}_i(\mathbf{r})|\mathbf{J}\rangle = \frac{1}{m}\hat{\rho}(\mathbf{r})\partial_i\hat{\theta}(\mathbf{r})|\mathbf{J}\rangle$. To implement the separation of charge and flux, we perform the point-splitting of the composite boson operator by the prescription

$$\hat{\phi}(\mathbf{r}) = e^{i\hat{\theta}(\mathbf{r}+\epsilon)}\sqrt{\hat{\rho}(\mathbf{r})}, \tag{48}$$

where the amplitude of the parameter $\epsilon$ will eventually be taken to zero. Here the charge is created at $\mathbf{r} + \epsilon$, while a thin flux tube is attached at the position of the current $\mathbf{r}$ since the CS gauge field couples to the density. The heuristics is that of a charged particle performing a cyclotron motion around the guiding center, to which is attached a CS flux tube. Note

though, that this is a mere interpretation of the above, mathematically precise, point-splitting prescription.

With this picture in mind, we generalize the phase transformation factor $\Phi_K$ in Eq. (35). In order to obtain the correct shift of the Jain state, we need to increase the orbital spin of the electrons in one of the groups, say the group $M_2$. There is no unique way of doing this since the phase transformation is obtained in a first quantized framework where there is no counterpart to the point-splitting (48). The simplest choice, expressed in terms of the phase transformation $\Phi_K$ in the multi-component case (35) with the $K$-matrix (47), is

$$
\begin{aligned}
\Phi_K^l(\{\mathbf{r}\}, \{\xi\}) &= \Phi_K(\{\mathbf{r}\}, \{\xi\}) \prod_{i \in M_2} \left(\frac{\epsilon_i}{\bar{\epsilon}_i}\right)^{-\frac{l}{2}} \\
&= \prod_{i<j \in M_1} \left(\frac{z_i - z_j}{\bar{z}_i - \bar{z}_j}\right)^{\frac{3}{2}} \prod_{i<j \in M_2} \left(\frac{\xi_i - \xi_j}{\bar{\xi}_i - \bar{\xi}_j}\right)^{\frac{3}{2}} \\
&\quad \times \prod_{\substack{i \in M_1 \\ j \in M_2}} \left(\frac{z_i - \xi_j}{\bar{z}_i - \bar{\xi}_j}\right)^{1} \prod_{i \in M_2} \left(\frac{\epsilon_i}{\bar{\epsilon}_i}\right)^{-\frac{l}{2}},
\end{aligned}
\tag{49}
$$

where $\xi = \mathbf{r} + \epsilon$. Here $(\xi, \bar{\xi})$ and $(\epsilon, \bar{\epsilon})$ denote the holomorphic and anti-holomorphic components of $\xi$ and $\epsilon$. Finally, $l$ is an integer which determines the orbital angular momentum, or orbital spin, of the particle at position $\xi$ around the flux at position $\mathbf{r}$. Strictly speaking, the above prescription, which is written entirely in terms of the positions of the charges, does not faithfully implement the idea of picking up phases from charges moving around fluxes, but it is correct in the limit $\epsilon_i \to 0$, as discussed in Appendix C.

The calculation of the ground state wave function then proceeds as in the previous section, taking the $K$ matrix (47). In particular, the Hamiltonian $H_B$ is as in Eq. (37) (with $\alpha = 1, 2$), and the composite boson wave function reads

$$
\Psi_B(\{\mathbf{r}\}, \{\xi\}) = \Big\langle \prod_{i \in M_1} e^{iQ_{1\beta}\hat{\theta}_\beta(\mathbf{r}_i)} \prod_{i \in M_2} e^{iQ_{2\beta}\hat{\theta}_\beta(\xi_i)} e^{-i\bar{\rho}_\alpha Q_{\alpha\beta} \int d^2 r \, \hat{\theta}_\beta(\mathbf{r})} \Big\rangle.
\tag{50}
$$

Multiplying with the phase factor (49), the fermionic wave function becomes

$$
\begin{aligned}
\Psi_F^{l,\{\epsilon\}}(\{z\}, \{\xi\}) &= \Big\langle \prod_{i \in M_1} e^{iQ_{1\beta}\hat{\phi}_\beta(z_i)} \prod_{j \in M_2} e^{iQ_{2\beta}\hat{\phi}_\beta(\xi_j)} \mathcal{O}_b \Big\rangle \prod_{i \in M_2} \left(\frac{\epsilon_i}{\bar{\epsilon}_i}\right)^{-\frac{l}{2}} \\
&= \prod_{i<j \in M_1} (z_i - z_j)^3 \prod_{i<j \in M_2} (z_i - z_j + \epsilon_i - \epsilon_j)^3 \\
&\quad \times \prod_{\substack{i \in M_1 \\ j \in M_2}} (z_i - z_j - \epsilon_j)^2 \prod_{i \in M_2} \left(\frac{\epsilon_i}{\bar{\epsilon}_i}\right)^{-\frac{l}{2}}.
\end{aligned}
$$

That this expression depends on the parameters $\epsilon$ is unsurprising, as these are a part of the singular gauge transformation. For fixed values of the parameters $\epsilon$, the gauge choice explicitly breaks rotational invariance. Since all gauge choices should be equally good, the simplest thing to do is to integrate over all directions of these vectors[4]. So we parametrize $\epsilon_i = |\epsilon_i| e^{i\vartheta_i}$ and integrate over all the angular variables.

In the present case, this prescription amounts to choosing $l = 1$, that is to increase the orbital spin of all particles in group $M_2$ by one before integrating over all the angular variables.

---

[4] To integrate over different gauge choices is a known technique in gauge theory, that is used e.g. to derive the gauge fixing term for general covariant gauges.

Since each $\epsilon_i$ gives a factor $e^{-i\vartheta_i}$, the angular integration only yields a non-vanishing result if the wave function contributes a compensating factor $e^{i\vartheta_i}$. By expanding the polynomials, this amounts to keeping $\epsilon_i$ to first order and is equivalent to taking a derivative of the $z's$ in the group $M_2$:

$$
\begin{aligned}
\Psi_F^{l=1}(\{z\}) &= \prod_{i\in M_2} \int_0^{2\pi} d\vartheta_i \, \Psi_F^{1,\{\epsilon\}}(z_1,\dots,z_N,\xi_1,\dots,\xi_N) \\
&= \prod_{i\in M_2} |\epsilon_i| \partial_{z_i} \prod_{i<j\in M_1} (z_i - z_j)^3 \prod_{i<j\in M_2} (z_i - z_j)^3 \prod_{\substack{i\in M_1 \\ j\in M_2}} (z_i - z_j)^2.
\end{aligned}
\tag{51}
$$

Renormalizing this expression to remove the factors $|\epsilon_i|$ and antisymmetrizing with respect to the two groups, this precisely yields the CF wave function for $\nu = 2/5$. Alternatively, we can define

$$
V_1(z) = e^{iQ_{1\beta}\hat{\phi}_\beta(z)}
$$
$$
V_2(z) = \partial_z e^{iQ_{2\beta}\hat{\phi}_\beta(z)},
$$

to obtain the following compact expression for the wave function

$$
\Psi_F(\{z\}) = \mathcal{A}\Big\langle \prod_{i=1}^N V_1(z_i) \prod_{i=N+1}^{2N} V_2(z_i) \mathcal{O}_b \Big\rangle.
\tag{52}
$$

Some technical comments on the result (52) might be useful:

1. There is no way to *deduce* the value of the orbital spin, neither of the ground state, given only the K-matrix, nor of excited states, given K and the l-vector. The orbital spin is an input in our theory. Here we have picked the lowest value for $l$ that will give an expression that does not vanish after antisymmetrization. Had we chosen a higher value, the wave function would contain higher order derivatives. The rationale for choosing the lowest $l$ is that adding "unnecessary" derivatives tends to move the electrons closer together (without changing the filling fraction) and is expected to increase the energy because of the repulsive interaction. Also, using the factor $(\epsilon_i/\bar{\epsilon}_i)^{l/2}$ is just one way to select the desired orbital spin. Using $\epsilon_i^l$ instead, for instance, would just amount to a renormalization of the wave function.

2. We note that the $\nu = 2/5$ wave function with $|M_1| = |M_2| = N$ is rather special. Namely, we can replace the factor $\prod_{i\in M_2} \partial_{z_i}$ by *any* set of $N$ derivatives without changing the resulting wave function by more than an overall factor. A proof of this rather surprising result, which applies to all wave functions in the $n = 2$ Jain series, is given in Appendix A. This means the above construction could be made much more 'symmetric', in the sense that we could have chosen to point split *all* particles in (49) and obtain the wave function by projecting to the lowest angular momentum state that survives antisymmetrization. We believe this also applies to more general states in the positive Jain series, i.e. with $n > 2$.

3. In this and in the following derivations, we do not specify the precise form of the background operator $\mathcal{O}_b$, but refer to the extensive discussion in [19, 40, 43].

4. We also ignored the corrections to the Gaussian factor coming from the background charge as shown in (16). The point-splitting will also affect the contribution to the correlator from the background charge, and expanding the point-split Gaussian factor to first order in $\epsilon$ yields $e^{-|z|^2/4}(1-\bar{z}\epsilon/4)$. This amounts to the substitution $\partial_z \to \partial_z - \bar{z}/4$, thus

giving a wave function with components in higher Landau levels. (There are also contributions in $\bar{\epsilon}$ that survive the angular integration, but those vanish in the limit $\varepsilon \to 0$.) The presence of $\bar{z}$'s can be dealt with in two ways, which are technically equivalent but conceptually rather different. The first one is to just project on the LLL, as is done in composite fermion calculations. This amounts to the replacement $\bar{z} \to 2\partial_z$, and thus to a trivial renormalization of the original wave function. The second way to arrive at the same result, which is described at the end of Sect. 5.3, is to interpret the coordinates $z$ and $\bar{z}$ not as position coordinates for the electrons, but as guiding center coordinates for their cyclotron motion [19,43]. This highlights that the mean-field treatment is not expected to capture the short-distance behavior of the wave function, which is closely connected to that our regularization procedure is not unique, as discussed above and in Appendix C.

### 5.2.2 The positive Jain series and the chiral hierarchy

We now generalize the above example for the $\nu = 2/5$ state to the leading positive Jain series. The states, at $\nu = \frac{n}{2n+1}$, and can be thought of in composite fermion language as resulting from filling $n$ CF Landau levels and attaching two vortices to each electron. They are characterized by the $n \times n$ $K$-matrix

$$
K = \begin{pmatrix} 3 & 2 & 2 & \dots & 2 \\ 2 & 3 & 2 & \dots & 2 \\ 2 & 2 & 3 & \cdots & 2 \\ \vdots & \vdots & & \ddots & \\ 2 & 2 & \cdots & 2 & 3 \end{pmatrix}.
\tag{53}
$$

The phase factor in Eq. (49) is generalized straightforwardly by replacing the coordinates $\mathbf{r}$ in $M_\alpha$ by $\xi = \mathbf{r} + \boldsymbol{\epsilon}^{(\alpha)}$, where $\boldsymbol{\epsilon}^{(\alpha)} = \varepsilon^{(\alpha)}(\cos\vartheta^{(\alpha)}, \sin\vartheta^{(\alpha)})$, where we set $\varepsilon^{(1)} = 0$. Thus,

$$
\Phi_K^{\{l\}}(\{\xi\}) = \Phi_K(\{\xi\}) \prod_\alpha \prod_{i \in M_\alpha} \left( \frac{\epsilon_i^{(\alpha)}}{\bar{\epsilon}_i^{(\alpha)}} \right)^{-l_\alpha/2},
\tag{54}
$$

where $\Phi_K$ is the phase factor Eq. (35) for the $K$-matrix (53). The corresponding wave functions, before the limits $\varepsilon^{(\alpha)} \to 0$ are taken, are straightforward to compute but the expressions are cumbersome and not very instructive. To extract the final wave functions, we follow the same strategy as in the previous section, i.e. we expand in powers of the $\varepsilon^{(\alpha)}$, and extract the leading term. For the states in the positive Jain series, this is rather easy: due to the symmetry between the different $\Lambda$-levels, precisely one extra derivative is needed for each new level in the hierarchy, i.e. $l_\alpha = \alpha - 1$. This yields the vertex operators,

$$
V_\alpha(z) = \partial_z^{\alpha-1} e^{iQ_{\alpha\beta}\hat{\phi}_\beta(z)},
\tag{55}
$$

in the CFT description, so that the fermionic wave function reads

$$
\Psi_F(\{z\}) = \mathcal{A} \left\langle \prod_{\alpha=1}^n \prod_{i \in M_\alpha} V_\alpha(z_i) \mathcal{O}_b \right\rangle.
\tag{56}
$$

More generally, hierarchy states obtained by condensing quasielectrons are not symmetric in the different components. A simple and experimentally relevant example is the $\nu = 4/11$ state, with the $K$-matrix

$$
K = \begin{pmatrix} 3 & 2 \\ 2 & 5 \end{pmatrix}.
\tag{57}
$$

In this case, one can show that antisymmetry requires derivatives on the second group, just as for $\nu = 2/5$. A natural generalization to a $n$-level state would be to take one extra derivative at each level[5] exactly as in the Jain series, yielding a CFT description with the same functional form as Eq. (56). This result follows if we take $l_\alpha = \alpha - 1$, and carry out the averages over the angles $\vartheta_i^{(\alpha)}$. We believe that the same result would follow even without taking averages, i.e. by a limiting procedure of the kind discussed in point 2 in the previous section. Such a general analysis is, however, technically challenging — the difficulty being to determine what terms survive the antisymmetrization — and we will not attempt to carry it out. In addition, the constants $l_\alpha$ are related to the spin vector by $2S_\alpha = K_{\alpha\alpha} + l_\alpha$, and can be chosen different from the minimal prescription, $S_\alpha = \alpha - 1$, needed for the wave function not to vanish identically. We have again not investigated how to define a limiting procedure that would properly define the corresponding higher spin vertex operators needed to obtain these more general wave functions.

We end this section with a general comment on the relation between the CFT expression for QH wave functions, and the GLCS expressions derived in this paper. As stressed by Wen, to describe a hierarchy state corresponding to an $n \times n$ K-matrix, one needs $n$ distinct electron operators [56], and in the previous sections we showed how, by a statistical transmutation implemented by $n$ gauge fields, we could recover a CFT representation of the wave functions. In particular, the holomorphic components of the $n$ phase fields $\theta_i$ emerged as the $n$ scalar fields needed to represent the $n$ electron operators. There is however a large freedom in representing a holomorphic wave function as a CFT correlator. As explained for instance in Ref. [39] the vertex operators for the electrons at level $n$ of the hierarchy can generally be written as

$$V_\alpha(z) =: \partial_z^{\alpha-1} e^{i \sum_\beta Q_{\alpha\beta} \hat{\phi}^\beta(z)} : \quad \alpha = 1, 2 \ldots n, \tag{58}$$

where $K = QQ^T$, with $Q$ a $n \times k$ matrix, with $k \geq n$. $V_\alpha$ has conformal spin $s_\alpha = \frac{1}{2}K_{\alpha\alpha} + \alpha - 1$. We can thus in general use more CFT bosons than the minimal number $n$ to express the electron operators at level $n$ in the hierarchy. The microscopic derivation singles out a minimal representation with $n$ fields, since they derive from the $n$ inequivalent electron operators.

## 5.3 GLCS theory for the negative Jain series

We now turn to states that result from condensations of quasiholes, as opposed to the chiral hierarchy obtained by only condensing quasielectrons. We begin by briefly reviewing the CFT descriptions of such states. In Refs. [19, 43] it was shown how to extend the CFT formalism to a general hierarchy state by including anti-holomorphic fields. Here, we show how these CFT expressions can be derived from a GLCS theory by generalizing the singular gauge transformation beyond the expression (54).

In the $K$-matrix formalism, a quasihole condensate amounts to a negative eigenvalue of the $K$-matrix, which translates into an anti-holomorphic Jastrow factor in the wave function and an associated anti-chiral edge mode. The procedure described in the previous section actually only works for positive definite $K$ matrices, and the way to proceed in the general case is to split the $K$-matrix as

$$K = \kappa - \bar{\kappa}, \tag{59}$$

where both $\kappa$ and $\bar{\kappa}$ are positive semi-definite. Writing $\kappa = qq^T$ and $\bar{\kappa} = \bar{q}\bar{q}^T$, the CFT description is in terms of the operators

$$V_\alpha(z, \bar{z}) =: \partial_z^{\sigma_\alpha} \partial_{\bar{z}}^{\bar{\sigma}_\alpha} e^{i \sum_\beta q_{\alpha\beta} \phi^\beta(z)} e^{i \sum_\beta \bar{q}_{\alpha\beta} \bar{\phi}^\beta(\bar{z})} :, \tag{60}$$

---

[5]This is a conjecture. For the Jain series, it can be shown that one derivative per level is the minimal prescription that ensures a non-vanishing wave function [40]. Numerical tests for small system sizes indicate that the same holds for more general states in the chiral hierarchy, but we have not managed to prove it.

that generalize (58). Here, the powers of derivatives are related to the spin vector. For more details, we refer to Refs. [19,39] where it is also discussed in some detail how that the resulting correlators, which are not holomorphic, should be interpreted as wave functions for the guiding center coordinates, and that the electronic wave functions are obtained by a convolution with a coherent state kernel, which effectively projects on the lowest Landau level.

We consider states in the negative Jain series $\nu = 2/3, 3/5, \ldots$ which are the particle-hole conjugates of those in the leading positive Jain series described above. In the composite fermion approach, these correspond to making a "reverse flux attachment" and we shall use a similar idea to generalize the statistical transmutation. As example, we take the $\nu = 2/3$ state with the $K$-matrix

$$K = \begin{pmatrix} 1 & 2 \\ 2 & 1 \end{pmatrix}.$$

This matrix is not positive definite, and to construct CFT wave functions, one must decompose it in a chiral and anti-chiral part [19]. This can be done in many ways, but in order to connect to composite bosons, it must be done in a way that can be represented with only two fields. This suggests the following decomposition [19]

$$K = \kappa - \bar{\kappa} = \frac{3}{2} \begin{pmatrix} 1 & 1 \\ 1 & 1 \end{pmatrix} - \frac{1}{2} \begin{pmatrix} 1 & -1 \\ -1 & 1 \end{pmatrix}. \tag{61}$$

The $\bar{\kappa}$ matrix has rank 1, which immediately suggests the following parametrization of the vertex operators

$$\begin{aligned} V_1(z, \bar{z}) &= e^{i\sqrt{\frac{3}{2}}\phi_1(z)} e^{i\bar{\phi}_2(\bar{z})/\sqrt{2}} \\ V_2(z, \bar{z}) &= \bar{\partial} e^{i\sqrt{\frac{3}{2}}\phi_1(z)} e^{-i\bar{\phi}_2(\bar{z})/\sqrt{2}}, \end{aligned} \tag{62}$$

and the corresponding wave function (before antisymmetrization) reads

$$\Psi_{2/3}(\{z\}) = \prod_{i<j\in M_1} |z_i - z_j|(z_i - z_j) \prod_{i\in M_2} \partial_{\bar{z}_i} \prod_{i<j\in M_2} |z_i - z_j|(z_i - z_j) \prod_{\substack{i\in M_1 \\ j\in M_2}} \frac{(z_i - z_j)^{\frac{3}{2}}}{(\bar{z}_i - \bar{z}_j)^{\frac{1}{2}}}. \tag{63}$$

Note that in spite of the fractional powers, this expression is single valued in all the coordinates, so the coherent state projection is well defined and will give a unique (although rather complicated) LLL wave function.

We turn to the GLCS theory for the $\nu = 2/3$ state. The decomposition (61) of the $K$ matrix suggests making the phase transformation $\Phi_K^l = \Phi_K \prod_i \left(\frac{\epsilon_i}{\bar{\epsilon}_i}\right)^{-l/2}$, with $\Phi_K = \Phi_\kappa \Phi_{\bar{\kappa}}^{-1}$. That is

$$\begin{aligned} \Phi_K^l(\{\mathbf{r}\}, \{\xi\}) = &\prod_{i<j\in M_1} \left(\frac{z_i - z_j}{\bar{z}_i - \bar{z}_j}\right)^{\frac{3}{4}} \left(\frac{\bar{z}_i - \bar{z}_j}{z_i - z_j}\right)^{\frac{1}{4}} \prod_{i<j\in M_2} \left(\frac{\xi_i - \xi_j}{\bar{\xi}_i - \bar{\xi}_j}\right)^{\frac{3}{4}} \left(\frac{\bar{\xi}_i - \bar{\xi}_j}{\xi_i - \xi_j}\right)^{\frac{1}{4}} \\ &\times \prod_{\substack{i\in M_1 \\ j\in M_2}} \left(\frac{z_i - \xi_j}{\bar{z}_i - \bar{\xi}_j}\right)^{\frac{3}{4}} \left(\frac{\bar{z}_i - \bar{\xi}_j}{z_i - \xi_j}\right)^{-\frac{1}{4}} \prod_{i\in M_2} \left(\frac{\epsilon_i}{\bar{\epsilon}_i}\right)^{-\frac{l}{2}}, \end{aligned}$$

where as before $\xi = \mathbf{r} + \epsilon$. This amounts to attaching flux tubes of strengths $\frac{3}{2}$ and $-\frac{1}{2}$, such that the associated statistical field strength is opposite the external magnetic field for the former, and in the same direction for the latter.

Following the steps that led from from (2) to (9), *mutatis mutandis*, the corresponding composite boson Hamiltonian becomes

$$H_B = \frac{1}{2m} \int d^2r \left[ \bar{\rho}_1 (\nabla \theta_1 - e\mathbf{A} + \mathbf{a} + \mathbf{c})^2 + \bar{\rho}_2 (\nabla \theta_2 - e\mathbf{A} + \mathbf{a} - \mathbf{c})^2 \right], \tag{64}$$

together with the constraints

$$b_{\mathbf{a}} = 2\pi \frac{3}{2}(\rho_1 + \rho_2) \equiv 2\pi \frac{3}{2}\rho$$
$$b_{\mathbf{c}} = 2\pi \frac{1}{2}(\rho_2 - \rho_1) \equiv 2\pi \frac{1}{2}\rho_{\mathbf{c}}.$$

The mean-field conditions are $\bar{b}_{\mathbf{a}} = eB$ and $\bar{b}_{\mathbf{c}} = 0$, and imply $\bar{\rho}_1 = \bar{\rho}_2 = \bar{\rho}/2$. The average density is fixed by $2\pi \frac{3}{2}\bar{\rho} = eB$. In the Coulomb gauge $\nabla \cdot \delta\mathbf{a} = \nabla \cdot \delta\mathbf{c} = 0$, we therefore find

$$H_B = \frac{1}{2m}\frac{\bar{\rho}}{2} \int d^2r \left[ \theta_1(-\nabla^2)\theta_1 + \theta_2(-\nabla^2)\theta_2 + 2\chi_{\mathbf{a}}(-\nabla^2)\chi_{\mathbf{a}} + 2\chi_{\mathbf{c}}(-\nabla^2)\chi_{\mathbf{c}} \right], \qquad (65)$$

where, as before, we have introduced the fields $\chi_{\mathbf{a},\mathbf{c}}$ given by $a^i = \epsilon^{ij}\partial_j\chi_{\mathbf{a}}$ and $c^i = \epsilon^{ij}\partial_j\chi_{\mathbf{c}}$. These are related to the new variables $\rho_{\mathbf{a}/\mathbf{c}}$ by $-\nabla^2\chi_{\mathbf{a}} = 2\pi\frac{3}{2}\rho$ and $-\nabla^2\chi_{\mathbf{c}} = 2\pi\frac{1}{2}\rho_{\mathbf{c}}$. Introducing angular variables conjugate to $\rho$ and $\rho_{\mathbf{c}}$ respectively

$$\theta_{\mathbf{a}} = \frac{1}{2}(\theta_1 + \theta_2)$$
$$\theta_{\mathbf{c}} = \frac{1}{2}(\theta_2 - \theta_1),$$

we find that the Hamilonian is the sum of two decoupled harmonic oscillators

$$\begin{aligned} H_B = \frac{\bar{\rho}}{2m} \int d^2r \Big[ &\theta_{\mathbf{a}}(\mathbf{r})(-\nabla^2)\theta_{\mathbf{a}}(\mathbf{r}) + \chi_{\mathbf{a}}(\mathbf{r})(-\nabla^2)\chi_{\mathbf{a}}(\mathbf{r}) \\ &+ \theta_{\mathbf{c}}(\mathbf{r})(-\nabla^2)\theta_{\mathbf{c}}(\mathbf{r}) + \chi_{\mathbf{c}}(\mathbf{r})(-\nabla^2)\chi_{\mathbf{c}}(\mathbf{r}) \Big]. \end{aligned} \qquad (66)$$

In the $\theta$-representation, the ground state wave functional is

$$\langle \theta(\mathbf{r})|\Psi\rangle = e^{\frac{1}{4\pi}\int d^2r\left[\frac{2}{3}\theta_{\mathbf{a}}(\mathbf{r})\nabla^2\theta_{\mathbf{a}}(\mathbf{r}) + 2\theta_{\mathbf{c}}(\mathbf{r})\nabla^2\theta_{\mathbf{c}}(\mathbf{r})\right]} e^{-i\bar{\rho}\int d^2r\,\theta_{\mathbf{a}}(\mathbf{r})}. \qquad (67)$$

Going to the path integral representation, using the operators $e^{i\theta_1(\mathbf{r})} = e^{i(\theta_{\mathbf{a}} - \theta_{\mathbf{c}})(\mathbf{r})}$ and $e^{i\theta_2(\mathbf{r})} = e^{i(\theta_{\mathbf{a}} + \theta_{\mathbf{c}})(\mathbf{r})}$, we get the composite boson wave function

$$\begin{aligned} \Psi_B(\mathbf{r}_1,\ldots,\mathbf{r}_N;\boldsymbol{\xi}_1,\ldots,\boldsymbol{\xi}_N) &= \left\langle e^{i(\theta_{\mathbf{a}} - \theta_{\mathbf{c}})(\mathbf{r}_1)} \cdots e^{i(\theta_{\mathbf{a}} - \theta_{\mathbf{c}})(\mathbf{r}_N)} e^{i(\theta_{\mathbf{a}} + \theta_{\mathbf{c}})(\boldsymbol{\xi}_1)} \cdots e^{i(\theta_{\mathbf{a}} + \theta_{\mathbf{c}})(\boldsymbol{\xi}_N)} e^{-i\bar{\rho}\int d^2r\,\theta_{\mathbf{a}}} \right\rangle \\ &= \prod_{i<j\in M_1} |\mathbf{r}_i - \mathbf{r}_j|^2 \prod_{i<j\in M_2} |\boldsymbol{\xi}_i - \boldsymbol{\xi}_j|^2 \prod_{i\in M_1, j\in M_2} |\mathbf{r}_i - \boldsymbol{\xi}_j|^1. \end{aligned} \qquad (68)$$

As before, we multiply by the phase factor $\Phi_K$ to obtain the fermionic wave function and integrate over the angles of the $\epsilon_i$, which amounts to taking derivatives in the second group. Going through this procedure directly, we obtain

$$\begin{aligned} \Psi_F(\{z,\bar{z}\},\{\xi,\bar{\xi}\}) &= \prod_{i\in M_2} \partial_{z_i} \prod_{i<j\in M_1} (z_i - z_j)|z_i - z_j| \prod_{i<j\in M_2} (z_i - z_j)|z_i - z_j| \\ &\times \prod_{i\in M_1, j\in M_2} (z_i - z_j)^{\frac{3}{2}}(\bar{z}_i - \bar{z}_j)^{-\frac{1}{2}}. \end{aligned} \qquad (69)$$

The groups each contain half of the particles since the densities are equal. If we write the phase factor as a ratio of correlators instead, we obtain from Eq. (68) the CFT representation

$$\begin{aligned} \Psi_F(\{z,\bar{z}\}) &= \prod_{i\in M_2} \partial_{z_i} \Big\langle \prod_{i\in M_1\cup M_2} e^{i\sqrt{\frac{3}{2}}\phi_{\mathbf{a}}(z_i)} \Big\rangle \Big\langle \prod_{i\in M_1} e^{-i\sqrt{\frac{1}{2}}\bar{\phi}_{\mathbf{c}}(\bar{z}_i)} \prod_{i\in M_2} e^{i\sqrt{\frac{1}{2}}\bar{\phi}_{\mathbf{c}}(\bar{z}_i)} \Big\rangle \\ &= \langle V_1(z_1,\bar{z}_1) \cdots V_1(z_N,\bar{z}_N) V_2(z_{N+1},\bar{z}_{N+1}) \cdots V_2(z_{2N},\bar{z}_{2N}) \mathcal{O}_b \rangle. \end{aligned} \qquad (70)$$

Here $\phi_{\mathbf{a}}(z) = \theta_{\mathbf{a}}(z) + \varphi_{\mathbf{a}}(z)$ and $\bar{\phi}_{\mathbf{c}}(\bar{z}) = \bar{\theta}_{\mathbf{c}}(\bar{z}) + \bar{\varphi}_{\mathbf{c}}(\bar{z})$ and we recognized the vertex operators from Eq. (62). Performing the antisymmetrization over the different groups, we precisely obtain (63), which after antisymmetrization and projection on the lowest Landau level, is the Jain wave function at $\nu = 2/3$.

It is fairly straightforward to generalize the above to the full negative Jain series, and in Appendix B we illustrate the general procedure with the $\nu = 3/5$ which is a level 3 state. We would assume that our method would also extend to the full hierarchy, that is including the mixed states, but we have not attempted to prove this.

We end this section with two comments:

1. We already stressed that to get a fermionic wave function for non-chiral states a convolution with a coherent state kernel is necessary, and that this amounts to a projection onto the lowest Landau level. To get composite fermion wave functions, such projections are needed already for the chiral states, since the unprojected functions reside in higher effective Landau levels and contain powers of $\bar{z}$. So the surprising fact is actually that the GLCS approach directly give holomorphic wave functions for the chiral hierarchy, although the electron mass remains as a parameter, and the quasiparticle excitation energies are not at the right scale. To put this unexpected success in perspective, we should remember that we only extracted the part of the wave function that is dominant at long distances, and there are sub-leading components in higher Landau levels [12]. Thus, if we were to calculate corrections by performing a derivative expansion, we would indeed have to project the result onto the lowest Landau level.

2. In Ref. [19] it was pointed out that although our $\Psi_{2/3}$ share the topological properties with the wave function obtained using reversed flux composite fermions, they differ in that the latter has an extra short distance repulsion factor $\sum_{i<j} |z_i - z_j|$. As first pointed out by Girvin and Jach in the context of the Laughlin wave functions [57], such factors can be introduced in any QH wave function without changing the topological properties [58]. These factors do not appear naturally, both in the composite fermion approach and the GLCS formalism derived in this paper. The most natural way to think of introducing such factors is to view them as part of the coherent state kernel that carries information about the short distance correlations that is not coded in the topological data.

# 6 Conclusion

In this paper we have obtained two main results. The first, concerning the Laughlin and general multi-component states, is a precise connection between the non-relativistic scalar fields appearing in the microscopic GLCS theory, and the relativistic scalar fields central to their CFT description.

The second, more important, result is a microscopic derivation of the CFT hierarchy wave functions starting from a multi-component GLCS theory. The derivation relies on a generalized statistical gauge transformation, based on a point-splitting between the flux and charge of the composite bosons. We find it quite satisfactory that we have managed to give a microscopic derivation of hierarchy wave functions that arguably is on par with the one for the Laughlin states.

We only briefly discussed quasihole excitations, but from the example we gave it is very likely that generalization to general hierarchy states will be rather straightforward. Constructing quasielectron excitations is more of a challenge, but we deem it possible using insights from Refs. [40, 52].

## Acknowledgements

We thank Eddy Ardonne for helpful discussions and comments on the manuscript. THH thanks Jon Magne Leinaas and Andrea Cappelli for discussions at an early stage of this work.

**Funding information**    MH and THH are supported by the Swedish Research Council. MH and YT are supported by the Knut and Alice Wallenberg foundation.

## A    Displaced derivative representation of Jain's wave function

Denote by $\Psi_{2/5}$ the composite fermion wave function obtained by filling two CF Landau levels and attaching two vortices, commonly written

$$\Psi_{2/5}(\{z\}) = \mathcal{P}_{\text{LLL}} \prod_{i<j} (z_i - z_j)^2 \Phi_{\nu^*=2}, \tag{71}$$

where $\Phi_{\nu^*=2}$ is the wave function for two filled CF Landau levels and $\mathcal{P}_{LLL}$ denotes the LLL projection. It has been shown [40] that this is identical to

$$\Psi_{2/5}(\{z\}) = \mathcal{A}\left\{ \partial_{z_{N+1}} \cdots \partial_{z_{2N}} \Psi^{(332)}(z_1, \ldots, z_N; z_{N+1}, \ldots, z_{2N}) \right\}, \tag{72}$$

where the $2N$ coordinates are divided into two groups $M_1, M_2$ of equal size, and where the (332) Halperin wave function reads

$$\Psi^{(332)}(z_1, \ldots, z_N; z_{N+1}, \ldots, z_{2N}) = \prod_{i<j\in M_1} (z_i - z_j)^3 \prod_{a<b\in M_2} (z_a - z_b)^3 \prod_{i,a\in M_1, M_2} (z_i - z_a)^2. \tag{73}$$

Acting with derivatives on the coordinates in the second group only, the antisymmetrization $\mathcal{A}$ sums over all distinct ways of dividing the particles over the two groups. From the perspective of the CF theory, the electrons in the second group are in the second CF Landau level and the derivatives result from the projection onto the lowest Landau level.

We show here that the wave function is unchanged – up to constants – if we act with any set of $N$ (distinct) derivatives, rather than derivatives of the second group only. To show this, we introduce an operator $e(i, j)$ which swaps the $i$-th and $j$-th coordinates in $M_1$ and $M_2$, respectively. We consider wave functions of the form

$$\Psi_{\{n\}}(\{z\}) \equiv \mathcal{A}\left\{ e(i_1, j_1) \cdots e(i_n, j_n)(\partial_{z_{N+1}} \cdots \partial_{z_{2N}})\Psi^{(332)}(\{z\}) \right\}, \tag{74}$$

where $i_1 \neq i_2 \neq \cdots \neq i_n$ and $j_1 \neq j_2 \neq \cdots \neq j_n$ and where the swap operators only act on the derivatives. First, we note that $\Psi_{\{n\}}$ is independent of the choice of the labels $\{i_k\}, \{j_k\}$: it only depends on the number of operators $n$, justifying the notation.

We make use of the Fock cyclic condition (see [59] and references therein) on the Halperin wave function

$$\Psi^{(332)}(z_1, \ldots, z_N; z_{N+1}, \ldots, z_{2N}) = \sum_{j=1}^{N} e(i, j) \Psi^{(332)}(z_1, \ldots, z_N; z_{N+1}, \ldots, z_{2N}), \tag{75}$$

which holds for any fixed $i$ in $M_1$. Applying this repeatedly for $i = 1, \ldots, n$, we find

$$\Psi_{2/5}(\{z\}) = \mathcal{A}\{\partial_{z_{N+1}} \cdots \partial_{z_{2N}} \sum_{j_1, \ldots, j_n} e(1, j_1) \cdots e(n, j_n) \Psi^{(332)}(\{z\})\}. \tag{76}$$

It follows that for a set of $j_1, \ldots, j_n$ where any $j_a = j_b$, the polynomial inside the antisymmetrization, including the derivative, is symmetric in the corresponding coordinates $z_a$ and $z_b$ (i.e. we act with $e(a, j_a)$ and $e(b, j_b) = e(b, j_a)$). Performing the antisymmetrization, such contributions vanish. We may therefore replace the sum by a sum over $j_1 \neq \cdots \neq j_n$. We find

$$
\begin{aligned}
\Psi_{2/5}(\{z\}) &= \sum_{j_1 \neq \cdots \neq j_n} \mathcal{A}\{\partial_{z_{N+1}} \cdots \partial_{z_{2N}} e(1, j_1) \cdots e(n, j_n) \Psi^{(332)}(\{z\})\} \\
&= (-1)^n \sum_{j_1 \neq \cdots \neq j_n} \mathcal{A}\{e(1, j_1) \cdots e(n, j_n)(\partial_{z_{N+1}} \cdots \partial_{z_{2N}})\Psi^{(332)}(\{z\})\}.
\end{aligned}
\tag{77}
$$

To obtain the second line, we view the antisymmetrization as a sum over permutations $\sigma$ and multiply each permutation on the left by the permutation $e(1, j_1) \cdots e(n, j_n)$. Since each swap is an odd permutation, this gives an overall factor $(-1)^n$. In the final expression, the operators act on the derivatives only, and we recognize the wave function $\Psi_{\{n\}}$. Using the aforementioned fact that they do not depend on the $\{j_k\}$, we obtain

$$
\Psi_{2/5}(\{z\}) = (-1)^n \binom{N}{n} \Psi_{\{n\}}(\{z\}).
\tag{78}
$$

Having worked out this example, we consider the more general wave functions in the Jain series with $\nu = \frac{n}{2pn+1}$. For $n = 2$ the CF wave functions are expressed in terms of two-component Halperin wave functions of type $(2p+1, 2p+1, 2p)$. Because the latter also obey the Fock cyclic condition, the argument holds for these CF wave functions as well.

For $n > 2$, CF wave functions are expressed in terms of $n$-component Halperin wave functions which obey a Fock cyclic condition for each pair of species. Although we have no general proof, we believe a similar argument applies to these wave functions as well.

# B  The $\nu = 3/5$ example

To illustrate that our approach to the $\nu = 2/3$ example in the main text generalizes, we work out the $n = 3$ case which corresponds to the $\nu = 3/5$ state. The $K$ matrix describing the state has the decomposition

$$
K = \begin{pmatrix} 1 & 2 & 2 \\ 2 & 1 & 2 \\ 2 & 2 & 1 \end{pmatrix} = \frac{5}{3} \begin{pmatrix} 1 & 1 & 1 \\ 1 & 1 & 1 \\ 1 & 1 & 1 \end{pmatrix} - \frac{1}{3} \begin{pmatrix} 2 & -1 & -1 \\ -1 & 2 & -1 \\ -1 & -1 & 2 \end{pmatrix}.
\tag{79}
$$

Turning to the GLCS theory, the phase transformation is a straightforward generalization of Eq. (64) and yields the Hamiltonian

$$
\begin{aligned}
H_B = \frac{1}{2m} \int d^2r \big[ &\bar{\rho}_1(\nabla\theta_1 - e\mathbf{A} + \mathbf{a} + \mathbf{c}_1 + \mathbf{c}_2)^2 + \bar{\rho}_2(\nabla\theta_2 - e\mathbf{A} + \mathbf{a} + \mathbf{c}_1 - \mathbf{c}_2)^2 \\
&+ \bar{\rho}_3(\nabla\theta_3 - e\mathbf{A} + \mathbf{a} - 2\mathbf{c}_1)^2 \big].
\end{aligned}
\tag{80}
$$

The gauge fields $\mathbf{a}, \mathbf{c}_1, \mathbf{c}_2$ are related to the original gauge fields by an orthogonal transformation that diagonalizes $K$. The constraints on the associated statistical field strengths are

$$
\begin{aligned}
b_a &= 2\pi \frac{5}{3}(\rho_1 + \rho_2 + \rho_3) = 2\pi \frac{5}{3}\rho \\
b_{c,1} &= -2\pi \frac{1}{6}(\rho_1 + \rho_2 - 2\rho_3) = -2\pi \frac{1}{6}\Delta_1 \\
b_{c,2} &= -2\pi \frac{1}{2}(\rho_1 - \rho_2) = -2\pi \frac{1}{2}\Delta_2.
\end{aligned}
\tag{81}
$$

The mean-field conditions fix $\bar{\rho}_1 = \bar{\rho}_2 = \bar{\rho}_3 = \bar{\rho}/3$, and $\bar{b}_{\mathbf{a}} = eB$ fixes the mean density $2\pi \frac{5}{3}\bar{\rho} = eB$. As a result, we find in the Coulomb gauge

$$H_B = \frac{1}{2m}\frac{\bar{\rho}}{3}\int d^2r\Big[\sum_{i=1}^{3}\theta_i(-\nabla^2)\theta_i + 3\chi_{\mathbf{a}}(-\nabla^2)\chi_{\mathbf{a}} + 6\chi_{\mathbf{c}_1}(-\nabla^2)\chi_{\mathbf{c}_1} + 2\chi_{\mathbf{c}_2}(-\nabla^2)\chi_{\mathbf{c}_2}\Big]. \quad (82)$$

Then, introducing angular fields canonically conjugate to $\rho, \Delta_1$ and $\Delta_2$;

$$\begin{aligned} \theta_{\mathbf{a}} &= \frac{1}{3}\theta_1 + \frac{1}{3}\theta_2 + \frac{1}{3}\theta_3 \\ \theta_{\mathbf{c}_1} &= \frac{1}{6}\theta_1 + \frac{1}{6}\theta_2 - \frac{1}{3}\theta_3 \\ \theta_{\mathbf{c}_2} &= \frac{1}{2}\theta_1 - \frac{1}{2}\theta_2, \end{aligned} \quad (83)$$

we find that the Hamiltonian is again a sum of decoupled harmonic oscillators

$$\begin{aligned} H_B &= \frac{\bar{\rho}}{2m}\int d^2r\big[\theta_{\mathbf{a}}(-\nabla^2)\theta_{\mathbf{a}} + \chi_{\mathbf{a}}(-\nabla^2)\chi_{\mathbf{a}}\big] + 2\frac{\bar{\rho}}{2m}\int d^2r\big[\theta_{\mathbf{c}_1}(-\nabla^2)\theta_{\mathbf{c}_1} + \chi_{\mathbf{c}_1}(-\nabla^2)\chi_{\mathbf{c}_1}\big] \\ &+ \frac{2}{3}\frac{\bar{\rho}}{2}\int d^2r\big[\theta_{\mathbf{c}_2}(-\nabla^2)\theta_{\mathbf{c}_2} + \chi_{\mathbf{c}_2}(-\nabla^2)\chi_{\mathbf{c}_2}\big]. \end{aligned} \quad (84)$$

## C  More on the singular phase transformations

As pointed out in the text, when the electrons are very close the point-splitting prescription (49), which involves only the position of the charges, does not properly describe charges encircling vortices. This can be achieved by taking seriously that in the charge vortex composite, the vortex is at the position $\mathbf{r}$ and the charge at position $\xi = \mathbf{r} + \epsilon$. Combining the result of taking a charge around a vortex and vice versa, we find the following phase transformation for the general two-component matrix $K = \begin{pmatrix} m & n \\ n & m \end{pmatrix}$:

$$\begin{aligned} \Phi_K^l &= \prod_{a<b\in M_1}\left(\frac{z_a - z_b}{\bar{z}_a - \bar{z}_b}\right)^{\frac{m}{2}} \prod_{a<b\in M_2}\left(\frac{z_a - z_b + \epsilon_a}{\bar{z}_a - \bar{z}_b + \bar{\epsilon}_a}\right)^{\frac{m}{4}}\left(\frac{z_a - z_b - \epsilon_b}{\bar{z}_a - \bar{z}_b - \bar{\epsilon}_b}\right)^{\frac{m}{4}} \\ &\times \prod_{\substack{a\in M_1 \\ b\in M_2}}\left(\frac{z_a - z_b}{\bar{z}_a - \bar{z}_b}\right)^{\frac{n}{4}} \prod_{\substack{a\in M_1 \\ b\in M_2}}\left(\frac{z_a - z_b - \epsilon_b}{\bar{z}_a - \bar{z}_b - \bar{\epsilon}_b}\right)^{\frac{n}{4}} \prod_{a\in M_2}\left(\frac{\epsilon_a}{\bar{\epsilon}_a}\right)^{-\frac{l}{2}}. \end{aligned} \quad (85)$$

As described in the text, we now fix the positions $z_a$ of the electrons and expand the above expression to leading order in the $\epsilon$ factors. This yields,

$$\begin{aligned} \Phi_K^l &= \prod_{a<b\in M_1}\left(\frac{z_a - z_b}{\bar{z}_a - \bar{z}_b}\right)^{\frac{m}{2}} \prod_{a<b\in M_2}\left(\frac{z_a - z_b + \epsilon_a/2 - \epsilon_b/2}{\bar{z}_a - \bar{z}_b + \bar{\epsilon}_a/2 - \bar{\epsilon}_b/2}\right)^{\frac{m}{2}} \\ &\times \prod_{\substack{a\in M_1 \\ b\in M_2}}\left(\frac{z_a - z_b - \epsilon_b/2}{\bar{z}_a - \bar{z}_b - \bar{\epsilon}_b/2}\right)^{\frac{n}{2}} \prod_{a\in M_2}\left(\frac{\epsilon_a}{\bar{\epsilon}_a}\right)^{-\frac{l}{2}}. \end{aligned} \quad (86)$$

The composite boson wave function is the same as in the main text,

$$\Psi_B = \prod_{a<b\in M_1}|\mathbf{r}_a - \mathbf{r}_b|^m \prod_{a<b\in M_2}|\mathbf{r}_a - \mathbf{r}_b + \epsilon_a - \epsilon_b|^m \prod_{a\in M_1, b\in M_2}|\mathbf{r}_a - \mathbf{r}_b - \epsilon_b|^n, \quad (87)$$

so multiplying with the phase factor (86) and again expanding to leading order in the $\epsilon$ factors and antisymmetrizing we finally get

$$
\begin{aligned}
\Psi_K^{l,\epsilon} = \mathcal{A} \prod_{a<b\in M_1} (z_b - z_a)^m \prod_{a<b\in M_2} \left( z_a - z_b + \frac{3}{4}\epsilon_i - \frac{3}{4}\epsilon_b \right)^m \\
\times \prod_{\substack{a\in M_1 \\ b\in M_2}} \left( z_a - z_b - \frac{3}{4}\epsilon_b \right)^n \prod_{a\in M_2} \left( \frac{\epsilon_a}{\bar{\epsilon}_a} \right)^{-\frac{l}{2}} ,
\end{aligned}
\tag{88}
$$

which is identical to (49), up to a trivial renormalization of the $\epsilon$ factors. We have also ignored the remaining factors in $\bar{\epsilon}$, which do not contribute when performing the integrals for $l > 0$ and in the limit $\epsilon \to 0$ (note that using the phase transformation (49) in the text, the anti-holomorphic parts cancel precisely). If we were to expand to non-leading orders, corresponding to choosing a non-minimal value for $l$ and a non-minimal shift, this would no longer be true. Instead it would correspond to letting the derivatives only act on parts of the Jastrow factors. These ambiguities, related to where in the holomorphic part of the wave function the derivatives are to act, only change the short-distance properties of the wave functions and can thus not change the topological properties.

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
