# Peer review of "Microscopic derivation of Ginzburg-Landau theories for hierarchical quantum Hall states"

_SciPost Physics, doi:SciPost Phys. 8, 079 (2020)_

## Round 1 · Referee Report · Anonymous (Referee 1) · 2020-3-12

Strengths

1-provides an explicit, microscopic and elegant connection between GLCS theories and CFT correlators for important quantum Hall state family
2-very pedagogical presentation

Weaknesses

1-introduction might further comment on the existing connections between GLCS theories and CFTs

Report

In their manuscript, the authors explore a microscopic construction scheme for fractional quantum Hall wave functions that connects Ginzburg-Landau-Chern-Simons theories and conformal field theory correlators. First applying their scheme to the Laughlin case, they show that, and how, their method works. The authors then find a natural extension of their scheme to multi-component Hall states, which in turn enables them to propose a scheme that allows to construct wave functions for hierarchical quantum Hall states. As the authors nicely explain, hierarchical states are in many ways less explored than other Hall states. The relevance of this work in my view mostly lies in providing an explicit connection between Ginzburg-Landau-Chern-Simons theories and conformal field theory correlators for hierarchical states, and thus shedding more light onto the latter.

The authors do a great job in summarizing the necessity and aim of their work, although the authors might further comment on the connections between GLCS theories and CFTs unearthed thus far. Their discussion of how their scheme applies to the Laughlin states is very pedagogical. The main result, in my view, is the generalization of their scheme to hierarchy states, a cornerstone of which is the correct description the orbital spin. I find that physical discussion relating point splitting and the orbital spin helpful, and the mathematical implementation elegant. If I wanted to critique the present work, the connection between the two feels not unappropriate, but a bit fuzzy. Still, I find the work overall very pedagogical, and recommend it for publication in SciPost.

A few minor remarks are in order:

1) There is a undefined reference to an equate on p. 14 (upper half).

2) I would enjoy an enlarged discussion of how the point splitting technically connects to Eq. (49) beyond the more technical Appendix C. With the current version, I understand that (49) expresses that the flux associated with particle A felt by particle B is spatially displaced from particle A’s charge position (and vice versa), which connects well with the point splitting argument above. It would be very enlightening if the authors could similarly extend their discussion of the factor (\epsilon_i/\bar{epsilon}_i)^l which is key to select the correct angular momentum „channel“. I see that this term gives the right math and is associated with the correct angular momentum, but is there a similarly simple explanation as for the displacements in the wavefunction implemented by \xi (together with the integration over phases, this factor looks somehow similar to a projection of the wave function onto the correct angle momentum channel)? As a related question, could the authors comment on how point splitting knows about the value of the angular momentum?

Requested changes

1-enlarge discussion of existing connections between GLCS theories and CFTs
2-if possible, comment on the selection of the correct angular momentum via (\epsilon_i/\bar{epsilon}_i)^l

  • validity: top
  • significance: high
  • originality: high
  • clarity: top
  • formatting: perfect
  • grammar: perfect

Author:  Yoran Tournois  on 2020-03-18  [id 767]

(in reply to Report 1 on 2020-03-12)

We are happy with the positive report from the referee, who clearly read our paper carefully. She/he has two suggestions for improvement, which we now address:

"1-enlarge discussion of existing connections between GLCS theories and CFTs"

To our knowledge the present paper is the first to make a direct connection between these approaches. Previously the connection has been only indirect, in that the same wave functions have been derived by two different methods. In particular, the GLCS theory has really only been used for the Laughlin states, although the generalization to the multicomponent case is rather trivial (we indeed looked for a good reference for this but without success). There is a work on a GLCS theory for the bosonic $\nu = 1$ nonabelian Moore-Read state, which also speculate about possible extensions to bosonic Read-Rezayi states. We do not know of any GLCS theory for the fermionic paired states. We have added a couple of sentences about this after the paragraph on page 4 ending with “both to their cyclotron motion in the lowest Landau level and their interaction [29–31].”

"2-if possible, comment on the selection of the correct angular momentum via $ (\epsilon_i/\bar{\epsilon}_i)^l$"

There is no way to deduce what is the orbital spin, either for the ground state, given only the K-matrix, or excited states, given the K-matrix and the l-vector. The spin vector $s$ is an input in our construction, and the factor $ (\epsilon_i/\bar{\epsilon}_i)^{l/2}$ is just one way to implement it. We could equally well have used, say, $\epsilon_i^l$ at the expense of having to renormalize the wave functions with powers of $|\epsilon_i|$. We have emphasized this point by extending the comment 1 on page 17.

---

## Round 1 · Referee Report · Anonymous (Referee 2) · 2020-4-19

Strengths

1 - Paper very well written
2 - Helpful bibliography

Report

This paper contains an interesting construction of GLCS theories and underlying CFTs. The paper should definitely be published, but I have a couple of small comments to make.

1 - The widely used commutation relation between density and phase, in Eq. (5), assumes that the phase is a hermitian operator. There are known difficulties in constructing such operators. Why are those difficulties not important here? Is this because only a continumm set of states is included, analogously to the case of position and momentum commutation relations?

2 - The authors mention in several points in the text singular gauge transformations and even include an appendix (C) on this. My question regards the phase $\theta$. The functional integral representation is assuming that $\theta$ runs in the interval $(-\infty,\infty)$. However, the phase is a periodic field, so I presume that the vortex fields are absorbed as a singular gauge transformation into the gauge field. Is this view correct?
  • validity: high
  • significance: high
  • originality: high
  • clarity: top
  • formatting: perfect
  • grammar: perfect

Author:  Yoran Tournois  on 2020-04-23  [id 806]

(in reply to Report 2 on 2020-04-19)

We are happy with the positive report from the second referee, who poses two questions which we now address.

  1. "The widely used commutation relation between density and phase, in Eq. (5), assumes that the phase is a hermitian operator. There are known difficulties in constructing such operators. Why are those difficulties not important here? Is this because only a continumm set of states is included, analogously to the case of position and momentum commutation relations?"

Strictly speaking it is indeed the case that the field $\theta$ is not Hermitian, and correspondingly that the operator $e^{i\theta}$ is not unitary. This can be traced back to that $e^{i\theta}$ annihilates the vacuum; thus the trouble comes from the fact that the number of particles cannot be less than zero. However, the operator $e^{i\theta}$ is well defined when acting on states with many particles which is the case we are interested in since we consider small density fluctuations $\delta \rho$ around a mean density $\bar{\rho}$. To clarify this point we have added the following comment below Eq. (5): "Note that, strictly speaking, the phase field $\theta$ is not Hermitian. However we are interested only small fluctuations around a mean density, and in this case $\theta$ can be effectively treated as Hermitian, and hence $e^{i\theta}$ as being unitary."

We add a reference (containing others references) as well.

  1. "The authors mention in several points in the text singular gauge transformations and even include an appendix (C) on this. My question regards the phase $\theta$. The functional integral representation is assuming that $\theta$ runs in the interval $(-\infty,\infty)$. However, the phase is a periodic field, so I presume that the vortex fields are absorbed as a singular gauge transformation into the gauge field. Is this view correct?"

The gauge transformations in question, such as in eq. (2) in the text and in appendix C, is singular since it changes the value of the phase factor $e^{\oint dx^i a_i}$ around the position of the particle. The variable $\theta$ introduced in the polar description on top of page 7, is an angular variable and remains so throughout, although a renormalization (as e.g. done just before Eq. (17) in the main text) is performed which changes its compactification radius. We never extend the integration range of $\theta$, since our final result is a standard compact scalar CFT. If we were to go from the GLCS theory to an effective topological field theory of the Wen-Zee type, we would divide the $\theta$ field into a regular and singular part, where the latter describes the vortex current. The regular part is then assumed to be a small perturbation so the integration range can be extended to $(-\infty,\infty)$ giving a delta function constraint in the path integral. It is an interesting problem to extend our formalism for the hierarchy to derive the correct Wen-Zee theory, but that is beyond the scope of this paper.

---

## Round 2 · List of Changes

- We added a comment on existing GLCS theories on page 4
- We added a comment on page 17, emphasizing that the orbital spin is an input to our construction to correctly implement the topological data
- We added a comment below Eq. 5 on the phase field
- We fixed a broken reference on page 13, a wrong variable in the mean-field solution below Eq. 6, and we added hats to fields in Eqs. (20),(28) and (58)
- We slightly modified the first footnote on page 8
- We added two references
- We fixed typos and grammar
- We reformulated the explanation of the status of the field $\hat{\phi}$ around eq. (20) and after eq. (46).

---

## Editorial Decision

published